# Fair Sortition Made Transparent

**Bailey Flanigan**
Department of Computer Science
Carnegie Mellon University
bflaniga@andrew.cmu.edu

**Gregory Kehne**
School of Engineering and Applied Sciences
Harvard University
gkehne@g.harvard.edu

**Ariel D. Procaccia**
School of Engineering and Applied Sciences
Harvard University
arielpro@seas.harvard.edu

## Abstract

Sortition is an age-old democratic paradigm, widely manifested today through the random selection of citizens' assemblies. Recently-deployed algorithms select assemblies *maximally fairly*, meaning that subject to demographic quotas, they give all potential participants as equal a chance as possible of being chosen. While these fairness gains can bolster the legitimacy of citizens' assemblies and facilitate their uptake, existing algorithms remain limited by their lack of transparency. To overcome this hurdle, in this work we focus on panel selection by uniform lottery, which is easy to realize in an observable way. By this approach, the final assembly is selected by uniformly sampling some pre-selected set of $m$ possible assemblies. We provide theoretical guarantees on the fairness attainable via this type of uniform lottery, as compared to the existing maximally fair but opaque algorithms, for two different fairness objectives. We complement these results with experiments on real-world instances that demonstrate the viability of the uniform lottery approach as a method of selecting assemblies both fairly and transparently.

## 1 Introduction

In a *citizens' assembly*, a panel of randomly chosen citizens is convened to deliberate and ultimately make recommendations on a policy issue. The defining aspect of citizens' assemblies is the randomness of the process, *sortition*, by which participants are chosen. In practice, the sortition process works as follows: first, volunteers are solicited via thousands of letters or phone calls, which target individuals chosen uniformly at random. Those who respond affirmatively form the *pool* of volunteers, from which a final panel will be chosen. Finally, a *selection algorithm* is used to randomly select some pre-specified number $k$ of pool members for the panel. To ensure adequate representation of demographic groups, the chosen panel is often constrained to satisfy some upper and lower quotas on feature categories such as age, gender, and ethnicity. We call a quota-satisfying panel of size $k$ a *feasible panel*. As this process illustrates, citizens' assemblies offer a way to involve the public in informed decision-making. This potential for civic participation has recently spurred a global resurgence in the popularity of citizens assemblies; they have been commissioned by governments and led to policy changes at the national level [19, 23, 12].

Prompted by the growing impact of citizens' assemblies, there has been a recent flurry of computer scientific research on sortition, and in particular, on the fairness of the procedure by which participants are chosen [2, 13, 12]. The most practicable result to date is a family of selection algorithms proposed by Flanigan et al. [12], which are distinguished from their predecessors by their use of randomness toward the goal of fairness: while previously-used algorithms selected pool members in

35th Conference on Neural Information Processing Systems (NeurIPS 2021).

a random but ad-hoc fashion, these new algorithms are *maximally fair*, ensuring that pool members have as equal probability as possible of being chosen for the panel, subject to the quotas.[1] To encompass the many interpretations of "as equal as possible," these algorithms permit the optimization of any fairness objective with certain convexity properties. There is now a publicly available implementation of the techniques of Flanigan et al. [12], called *Panelot*, which optimizes the egalitarian notion that no pool member has too little selection probability via the *Leximin* objective from fair division [21, 14]. This algorithm has already been deployed by several groups of panel organizers, and has been used to select dozens of panels worldwide.

Fairness gains in the panel selection process can lend legitimacy to citizens' assemblies and potentially increase their adoption, but only insofar as the public trusts that these gains are truly realized. Currently, the potential for public trust in the panel selection process is limited by multiple factors. First, the latest panel selection algorithms select the final panel via behind-the-scenes computation. When panels are selected in this manner, observers cannot even verify that any given pool member has *any* chance of being chosen for the panel. A second and more fundamental hurdle is that randomness and probability, which are central to the sortition process, have been shown in many contexts to be difficult for people to understand and reason about [24, 20, 28]. Aiming to address these shortcomings, we propose and pursue the following notion of transparency in panel selection:

> *Transparency:* Observers should be able to, without reasoning in-depth about probability, (1) understand the probabilities with which each individual will be chosen for the panel *in theory*, and (2) verify that individuals are actually selected with these probabilities *in practice*.

In this paper, we aim to achieve transparency and fairness simultaneously: this means advancing the defined goal of transparency, while preserving the fairness gains obtained by maximally fair selection algorithms. Although this task is reminiscent of existing AI research on trade-offs between fairness or transparency with other desirable objectives [4, 11, 3, 27], to our knowledge, this is the first investigation of the trade-off between fairness and transparency.

Setting aside for a moment the goal of fairness, we consider a method of random decision-making that is already common in the public sphere: the uniform lottery. To satisfy quotas, a uniform lottery for sortition must randomize not over individuals, but over entire feasible panels. In fact, this approach has been suggested by practitioners, and was even used in 2020 to select a citizens' assembly in Michigan. The following example, which closely mirrors that real-world pilot,[2] illustrates that panel selection via uniform lottery is naturally consistent with the transparency notion we pursue.

Suppose we construct 1000 feasible panels from a pool (possibly with duplicates), numbered 000-999, and publish an (anonymized) list of which pool members are on each panel. We then inform spectators that we will choose each panel with equal probability. This satisfies criterion (1): spectators can easily understand that all panels will be chosen with the same probability of 1/1000, and can easily determine each individual's selection probability by counting the number of panels containing the individual. To satisfy criterion (2), we enact the lottery by drawing each of the three digits of the final panel number individually from lottery machines. Lottery spectators can confirm that each ball is drawn with equal probability; this provides confirmation that panels are indeed being chosen with uniform probabilities, thus confirming the enactment of the proposed individual selection probabilities. In addition to its conventionality as a source of randomness, decision-making via drawing lottery balls invites an exciting spectacle, which can promote engagement with citizens' assemblies.

This simple method neatly satisfies our transparency criteria, but it has one obvious downside: a uniform lottery over an arbitrary set of feasible panels does not guarantee any measure of equal probabilities to individuals. In fact, it is not even clear that the *fairest possible* uniform lottery over $m$ panels, where $m$ is a number conducive to selection by physical lottery (e.g. $m = 1000$), would not be significantly less fair than maximally fair algorithms, which sample the fairest possible unconstrained distribution over panels. For example, if $m$ is too small, there may be *no* uniform lottery which gives all individuals non-zero selection probability, even if each individual appears

---

[1]Quotas can preclude giving individuals exactly equal probabilities: if the panel must be 1/2 men, 1/2 women but the pool is split 3/4 men, 1/4 women, then some women must be chosen more often than some men.

[2]Of By For's pilot of live panel selection via lottery can be viewed at `https://vimeo.com/458304880#t=17m59s` from 17:59 to 21:23. For a more detailed description, see Figure 3 and surrounding text in [12].

on some feasible panel (and so can attain a non-zero selection probability under an unconstrained distribution).

Fortunately, empirical evidence suggests that there is hope: in the 2020 pilot mentioned above, a uniform lottery over $m =1000$ panels was found that nearly matched the fairness of the maximally fair distribution generated by Panelot. Motivated by this anecdotal evidence, we aim to understand whether such a fair uniform lottery is guaranteed to exist in general, and if it does, how to find it. We summarize this goal in the following research questions:

> *Does there exist a uniform lottery over $m$ panels that nearly preserves the fairness of the maximally fair unconstrained distribution over panels?* And,
> *Algorithmically, how do we compute such a uniform lottery?*

**Results and Contributions.** After describing the model in Section 2, in Section 3 we prove that it is possible to round an (essentially) arbitrary distribution over panels to a uniform lottery while preserving *all* individuals' selection probabilities up to only a small bounded deviation. These results use tools from discrepancy theory and randomized rounding. Intuitively, this bounded change in selection probabilities implies bounded losses in fairness; we formalize this intuition in Section 4, showing that there exists in general a uniform lottery that is nearly maximally fair, with respect to multiple choices of fairness objective. Although we would ideally like to give such bounds for the *Leximin* fairness objective, due to its use practice, we cannot succinctly represent bounds for this objective because it is not scalar valued. We therefore give bounds for *Maximin*, a closely related egalitarian objective which only considers the minimum selection probability given to any pool member [7]. We discuss in Section 4 why bounds on loss in Maximin fairness are, in the most meaningful sense, also bounds on loss in Leximin fairness. We additionally give upper bounds on the loss in *Nash Welfare* [21], a similarly well-established fairness objective that has also been implemented in panel selection tools [18].

Finally, in Section 5, we consider the algorithmic question in practice: given a maximally fair distribution over panels, can we actually *find* nearly maximally fair uniform lotteries that match our theoretical guarantees? To answer this question, we implement two standard rounding algorithms, along with near-optimal (but more computationally intensive) integer programming methods, for finding uniform lotteries. We then evaluate the performance of these algorithms in 11 real-world panel selection instances. We find that in all instances, we can compute uniform lotteries that nearly exactly preserve not only fairness with respect to both objectives, but *entire sets* of Leximin-optimal marginals, meaning that from the perspective of individuals, there is essentially no difference between using a uniform lottery versus the optimal unconstrained distribution sampled by the latest algorithms. We discuss these results, their implications, and how they can be deployed directly into the existing panel selection pipeline in Section 6.

## 2 Model

**Panel Selection Problem.** First, we formally define the task of panel selection for citizens' assemblies. Let $N = [n]$ be the *pool* of volunteers for the panel—individuals from the population who have indicated their willingness to participate in response to an invitation. Let $F = \{f_t\}_t$ denote a fixed set of *features* of interest. Each feature $f_t : N \to \Omega_t$ maps each pool member to their value of that feature, where $\Omega_t$ is the set of $f_t$'s possible values. For example, for feature $f_t =$ "gender", we might have $\Omega_t = \{$"male","female", "non-binary"$\}$. We define individual $i$'s *feature vector* $F(i) = (f_t(i))_t \in \prod_t \Omega_t$ to be the vector encoding their values for all features in $F$.

As is done in practice and in previous research [13, 12], we impose that the chosen panel $P$ must be a subset of the pool of size $k$, and must be representative of the broader population with respect to the features in $F$. This representativeness is imposed via *quotas:* for each feature $f$ and corresponding value $v \in \Omega$, we may have lower and upper quotas $l_{f,v}$ and $u_{f,v}$. These quotas require that the panel contain between $l_{f,v}$ and $u_{f,v}$ individuals $i$ such that $f(i) = v$.

In terms of these parameters, we define an instance of the panel selection problem as: given $(N, k, F, l, u)$—a pool, panel size, set of features, and sets of lower and upper quotas—randomly select a *feasible panel*, where a feasible panel is any set of individuals $P$ from the collection $\mathcal{K}$:

$$\mathcal{K} := \left\{ P \in \binom{N}{k} : l_{f,v} \le |\{i \in P : f(i) = v\}| \le u_{f,v} \text{ for all } f, v \right\}.$$

**Maximally Fair Selection Algorithms.** A *selection algorithm* is a procedure that solves instances of the panel selection problem. A selection algorithm's level of fairness on a given instance is determined by its *panel distribution* $p$, the (possibly implicit) distribution over $\mathcal{K}$ from which it draws the final panel. Because we care about fairness to individual pool members, we evaluate the fairness of $p$ in terms of the fairness of selection probabilities, or *marginals*, that $p$ implies for all pool members.[3] We denote the vector of marginals implied by $p$ as $\pi$, and we will sometimes specify a panel distribution as $p, \pi$ to explicitly denote this pair. We say that $\pi$ is *realizable* if it is implied by some distribution $p$ over the feasible panels $\mathcal{K}$.

*Maximally fair* selection algorithms are those which solve the panel selection problem by sampling a specifically chosen $p$: one which implies marginals $\pi$ that allocate probability as fairly as possible across pool members. The fairness of $p, \pi$ is measured by a *fairness objective* $\mathcal{F}$, which maps an allocation—in this case, of selection probability to pool members—to a real number measuring the allocation's fairness. Fixing an instance, a fairness objective $\mathcal{F}$, and a panel distribution $p$, we express the fairness of $p$ as $\mathcal{F}(p)$. Existing maximally fair selection algorithms can maximize a wide range of fairness objectives, including those considered in this paper.

**Leximin, Maximin, and Nash Welfare.** Of the three fairness objectives we consider in this paper, Maximin and Nash Welfare (NW) have succinct formulae. For $p, \pi$ they are defined as follows, where $\pi_i$ is the marginal of individual $i$:

$$\text{Maximin}(p) := \min_{i \in N} \pi_i, \qquad \text{NW}(p) := \left( \prod_i \pi_i \right)^{1/n}.$$

Intuitively, NW maximizes the geometric mean, prioritizing the marginal $\pi_i$ of each individual $i$ in proportion to $\pi_i^{-1}$. Maximin maximizes the marginal probability of the individual least likely to be selected. Finally, Leximin is a refinement of Maximin, and is defined by the following algorithm: first, optimize Maximin; then, fixing the minimum marginal as a lower bound on any marginal, maximize the second-lowest marginal; and so on.

**Our task: quantize a maximally fair panel distribution with minimal fairness loss.** We define a $1/m$-*quantized* panel distribution as a distribution over all feasible panels $\mathcal{K}$ in which all probabilities are integer multiples of $1/m$. We use $\bar{p}$ to denote a panel distribution with this property. Formally, while an (unconstrained) panel distribution $p$ lies in $\mathcal{D} := \{p \in \mathbb{R}_+^{|\mathcal{K}|} : \|p\|_1 = 1\}$, a $1/m$-quantized panel distribution in $\bar{p}$ lies in $\overline{\mathcal{D}} := \{\bar{p} \in (\mathbb{Z}_+/m)^{|\mathcal{K}|} : \|\bar{p}\|_1 = 1\}$. Note that a $1/m$-quantized distribution $\bar{p}$ immediately translates to a physical uniform lottery of over $m$ panels (with duplicates): if $\bar{p}$ assigns probability $\ell/m$ to panel $P$, then the corresponding physical uniform lottery would contain $\ell$ duplicates of $P$. Thus, if we can compute a $1/m$-quantized panel distribution $\bar{p}$ with fairness $\mathcal{F}(\bar{p})$, then we have designed a physical uniform lottery over $m$ panels with that same level of fairness.

Our goal follows directly from this observation: we want to show that given an instance and desired lottery size $m$, we can compute a $1/m$-quantized distribution $\bar{p}$ that is nearly as fair, with respect to a fairness notion $\mathcal{F}$, as the maximally fair panel distribution in this instance $p^* \in \arg\max_{p \in \mathcal{D}} \mathcal{F}(p)$. We define the *fairness loss* in this quantization process to be the difference $\mathcal{F}(p^*) - \mathcal{F}(\bar{p})$. We are aided in this task by the existence of practical algorithms for computing $p^*$ Flanigan et al. [12], which allows us to use $p^*$ as an input to the quantization procedure we hope to design. For intuition, we illustrate this quantization task in Figure 1, where $\pi^*, \bar{\pi}$ are the marginals implied by $p^*, \bar{p}$, respectively. Since the fairness of $p^*, \bar{p}$ are computed in terms of $\pi^*, \bar{\pi}$, it is intuitive that a quantization process that results in small *marginal discrepancy*, defined as the maximum change in any marginal $\|\pi - \bar{\pi}\|_\infty$, should also have small fairness loss. This idea motivates the upcoming section, in which we give quantization procedures with provably bounded marginal discrepancy, forming the foundation for our later bounds on fairness loss.

---

[3] A panel distribution $p$ implies a unique vector of marginals $\pi$ as follows: fixing $p, \pi$, a pool member $i$'s marginal selection probability $\pi_i$ is equal to the probability of drawing a panel from $p$ containing that pool member. For a more detailed introduction to the connection between panel distributions and marginals, we refer readers to Flanigan et al. [12].

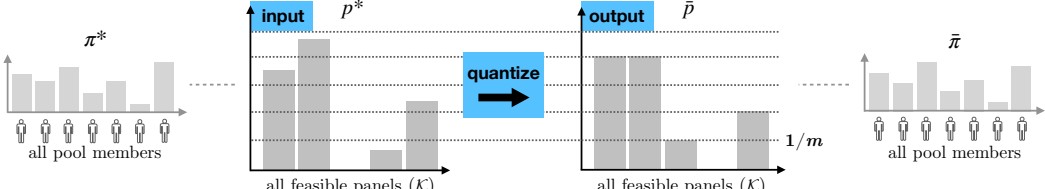

Figure 1: The quantization task takes as input a maximally fair panel distribution $p^*$ (implying marginals $\pi^*$), and outputs a $1/m$-quantized panel distribution $\bar{p}$ (implying marginals $\bar{\pi}$).

## 3 Theoretical Bounds on Marginal Discrepancy

Here we prove that for a fixed panel distribution $p, \pi$, there exists a uniform lottery $\bar{p}, \bar{\pi}$ such that $\|\pi - \bar{\pi}\|_\infty$ is bounded. Preliminarily, we note that it is intuitive that bounds on this discrepancy should approach 0 as $m$ becomes large with respect to $n$ and $k$. To see why, begin by fixing some distribution $p, \pi$ over panels: as $m$ becomes large, we approach the scenario in which a uniform lottery $\bar{p}$ can assign panels arbitrary probabilities, providing increasingly close approximations to $p$. Since the marginals $\pi_i$ are continuous with respect to $p$, as $\bar{p} \to p$ we have that $\bar{\pi}_i \to \pi_i$ for all $i$.

While this argument demonstrates convergence, it provides neither efficient algorithms nor tight bounds on the rate of convergence. In this section, our task is therefore to bound the rate of this convergence as a function of $m$ and the other parameters of the instance. All omitted proofs of results from this section are included in Appendix B.

### 3.1 Worst-Case Upper Bounds

Our first set of upper bounds result from rounding STANDARD LP, the LP that most directly arises from our problem. This LP is defined in terms of a panel distribution $p, \pi$, and $M$, an $n \times |\mathcal{K}|$ matrix describing which individuals are on which feasible panels: $M_{i,P} = 1$ if $i \in P$ and $M_{i,P} = 0$ otherwise.

STANDARD LP
$$Mp = \pi \tag{3.1}$$
$$\|p\|_1 = 1 \tag{3.2}$$
$$p \geq 0.$$

Here, (3.1) specifies $n$ total constraints. Our goal is to round $p$ to a uniform lottery $\bar{p}$ over $m$ panels (so the entries $\bar{p}$ are multiples of $1/m$) such that (3.2) is maintained exactly, and no constraint in (3.1) is relaxed by too much, i.e., $\|Mp - M\bar{p}\|_\infty = \|\pi - \bar{\pi}\|_\infty$ remains small.

Randomized rounding is a natural first approach. Any randomized rounding scheme satisfying negative association (which includes several that respect (3.2)) yields the following bound:

**Theorem 3.1.** *For any realizable $\pi$, we may efficiently randomly generate $\bar{p}$ such that its marginals $\bar{\pi}$ satisfy*

$$\|\pi - \bar{\pi}\|_\infty = O\left(\frac{\sqrt{n \log n}}{m}\right).$$

Fortunately, there is potential for improvement: randomized rounding does not make full use of the fact that $M$ is $k$-column sparse, due to each panel in $\mathcal{K}$ containing exactly $k$ individuals. We use this sparsity to get a stronger bound when $n \gg k^2$, which is a practically significant parameter regime. The proof applies a dependent rounding algorithm based on a theorem of Beck and Fiala [1], to which a modification ensures the exact satisfaction of constraint (3.2).

**Theorem 3.2.** *For any realizable $\pi$, we may efficiently construct $\bar{p}$ such that its marginals $\bar{\pi}$ satisfy*

$$\|\pi - \bar{\pi}\|_\infty \leq k/m.$$

This bound is already meaningful in practice, where $k \ll m$ is insured by the fact that $m$ is pre-chosen along with $k$ prior to panel selection. Note also that $k$ is typically on the order of 100

(Table 1), whereas a uniform lottery can in practice be easily made orders of magnitude larger, as each additional factor of 10 in the size of the uniform lottery requires drawing only one more ball (and there is no fairness cost to drawing a larger lottery, since increasing $m$ allows for uniform lotteries which better approximate the unconstrained optimal distribution).

## 3.2 Beyond-Worst-Case Upper Bounds

As we will demonstrate in Section 3.3, we cannot hope for a better worst-case upper bound than $\text{poly}(k)/m$. We thus shift our consideration to instances which are "simple" in their feature structure, having a small number of features (Theorem B.7), a limited number of unique feature vectors in the pool (Theorem 3.3), or multiple individuals that share each feature vector present (Theorem B.8). The beyond-worst-case bounds given by Theorem 3.3 and Theorem B.8 asymptotically dominate our worst-case bounds in Theorem 3.1 and Theorem 3.2, respectively. Moreover, Theorem 3.3 dominates all other upper bounds in 10 of the 11 practical instances studied in Section 5.

We note that while our worst-case upper bounds implied the near-preservation of *any* realizable set of marginals $\pi$, some of our beyond-worst-case results apply to only realizable $\pi$ which are *anonymous*, meaning that $\pi_i$ are equal for all $i$ with equal feature vectors. We contend that any reasonable set of marginals should have this property,[4] and furthermore that the "anonymization" of any realizable $\pi$ is also realizable (Claim B.6); hence this restriction is insignificant. Our beyond-worst-case bounds also differ from our worst-case bounds in that they depart from the paradigm of rounding $p$, instead randomizing over panels that may fall outside the support of $p$.

The main beyond-worst-case bound we give, stated below, is parameterized by $|\mathcal{C}|$, where $\mathcal{C}$ is the set of unique feature vectors that appear in the pool. All omitted proofs and other beyond worst-case results are stated and proven in Appendix B.

**Theorem 3.3.** *If $\pi$ is anonymous and realizable, then we may efficiently construct $\bar{p}$ such that its marginals $\bar{\pi}$ satisfy*

$$\|\pi - \bar{\pi}\|_\infty = O\left(\frac{\sqrt{|\mathcal{C}| \log |\mathcal{C}|}}{m}\right).$$

$|\mathcal{C}|$ is at most $n$, so this bound dominates Theorem 3.1. In 10 of the 11 real-world instances we study, $|\mathcal{C}|$ is also smaller than $k^2$ (Appendix A), in which case this bound also dominates Theorem 3.2.

At a high level, our beyond-worst-case upper bounds are obtained not by directly rounding $p$, but instead using the structure of the sortition instance to abstract the problem into one about "types." For this bound we then solve an LP in terms of "types," round that LP, and then reconstruct a rounded panel distribution $\bar{p}, \bar{\pi}$ from the "type" solution. In particular, the *types* of individuals are the feature vectors which appear in the pool, and *types* of panels are the multisets of $k$ feature vectors that satisfy the instance quotas. Fixing an instance, we project some $p$ into type space by viewing it as a distribution $\mathfrak{p}$ over types of panels $\mathfrak{K}$, inducing marginals $\tau_c$ for each type individuals $c \in \mathcal{C}$.

To begin, we define the TYPE LP, which is analogous to Eq. (3.1). We let $Q$ be the type analog of $M$, so that entry $Q_{cj}$ is the number of individuals $i$ with $F(i) = c$ contained in panels of type $j \in \mathfrak{K}$.[5] Then,

$$\text{TYPE LP} \qquad\qquad Q\,\mathfrak{p} = \tau \qquad\qquad\qquad (3.3)$$
$$\|\mathfrak{p}\|_1 = 1 \qquad\qquad\qquad (3.4)$$
$$\mathfrak{p} \geq 0.$$

We round $\mathfrak{p}$ in this LP to a panel type distribution $\bar{\mathfrak{p}}$ while preserving (3.4). All that remains, then, is to construct some $\bar{p}, \bar{\pi}$ such that $p$ is consistent with $\bar{\mathfrak{p}}$ and $\|\pi - \bar{\pi}\|_\infty$ is small. This $\bar{p}$ is in general supported by panels outside of $supp(p)$, unlike the $\bar{p}$ obtained by Theorem 3.1. It is the anonymity of $\pi$ which allows us to construct these new panels and prove that they are feasible for the instance.

---

[4]The class of all anonymous marginals $\pi$ includes the maximizers $\pi^*$ of all reasonable fairness objectives, and second, this condition is satisfied by all existing selection algorithms used in practice, to our knowledge.

[5]Completing the analogy, $\mathcal{C}, \mathfrak{K}, Q, \mathfrak{p}, \bar{\mathfrak{p}}, \tau$ are the "type" versions of $N, \mathcal{K}, M, p, \bar{p}, \pi$ from the original LP.

## 3.3 Lower Bounds

This method of using bounded discrepancy to derive nearly fairness-optimal uniform lotteries has its limits, since there are even sparse $M$ and fractional $x$ for which no integer $\bar{x}$ yields nearby $M\bar{x}$. In the worst case, we establish lower bounds by modifying those of Beck and Fiala [25]:

**Theorem 3.4.** *There exist $p, \pi$ for which for all uniform lotteries $\bar{p}, \bar{\pi}$,*

$$\min_{\bar{p} \in \mathcal{D}} \|\pi - \bar{\pi}\|_\infty = \Omega\left(\frac{\sqrt{k}}{m}\right).$$

Our $k$-dependent upper and lower bounds are separated by a factor of $\sqrt{k}$, matching the current upper and lower bounds of the Beck-Fiala conjecture as applied to linear discrepancy (also known as the lattice approximation problem [26]). The respective gaps are incomparable, however, since for a given $x \in [0,1]^n$, the former problem aims to minimize $\|M(x - \bar{x})\|_\infty$ over $\bar{x} \in \{0,1\}^n$, while we aim to do the same over a subset of the $\bar{x} \in \mathbb{Z}^n$ for which $\sum_j x_j = \sum_j \bar{x}_j$ (see Lemma B.4).

# 4 Theoretical Bounds on Fairness Loss

Since the fairness of a distribution $p$ is determined by its marginals $\pi$, it is intuitive that if uniform lotteries incur only small marginal discrepancy (per Section 3), then they should also incur only small fairness losses. This should hold for any fairness notion that is sufficiently "smooth" (i.e., doesn't change too quickly with changing marginals) in the vicinity of $p, \pi$.

Although our bounds from Section 3 apply to any reasonable initial distribution $p$, we are particularly concerned with bounding fairness loss from *maximally fair* initial distributions $p^*$. Here, we specifically consider such $p^*$ that are optimal with respect to Maximin and NW. We note that, since there exist anonymous $p^*, \pi^*$ that maximize these objectives, we can apply any upper bound from Section 3 to upper bound $\|\pi^* - \bar{\pi}\|_\infty$. We defer omitted proofs to Appendix C.

## 4.1 Maximin

Since Leximin is the fairness objective optimized by the maximally fair algorithm used in practice, it would be most natural to start with a $p^*$ that is Leximin-optimal and bound fairness loss with respect to this objective. However, the fact that Leximin fairness cannot be represented by a single scalar value prevents us from formulating such an approximation guarantee. Instead, we first pursue bounds on the closely-related objective, Maximin. We argue that in the most meaningful sense, a worst-case Maximin guarantee *is* a Leximin guarantee: such a bound would show limited loss in the minimum marginal, and it is Leximin's *lexicographically first priority* to maximize the minimum marginal.

First, we show there exists some $\bar{p}, \bar{\pi}$ that gives bounded Maximin loss from $p^*, \pi^*$, the Maximin-optimal unconstrained distribution. This bound follows from Theorems 3.3 and B.8, using the simple observation that $\bar{p}$ can decrease the lowest marginal given by $p^*$ by no more than $\|\pi^* - \bar{\pi}\|_\infty$. Here $n_{min} := \min_c n_c$ denotes the smallest number of individuals which share any feature vector $c \in \mathcal{C}$.

**Corollary 4.1.** *By Theorem 3.3 and B.8, for Maximin-optimal $p^*$, there exists a uniform lottery $\bar{p}$ that satisfies*

$$\mathrm{Maximin}(p^*) - \mathrm{Maximin}(\bar{p}) = \frac{1}{m} \cdot O\left(\min\left\{\sqrt{|\mathcal{C}| \log |\mathcal{C}|}, \ \frac{k}{n_{min}} + 1\right\}\right).$$

Theorem 3.4 demonstrates that we cannot get an upper bound on Maxmin loss stronger than $O(\sqrt{k}/m)$ using a uniform bound on changes in all $\pi_i$. However, since Maximin is concerned only with the smallest $\pi_i$, it seems plausible that better upper bounds on Maximin loss could result from rounding $\pi$ while tightly controlling only losses in the smallest $\pi_i$'s, while giving freer reign to larger marginals. We show that this is not the case by further modifying the instances from Theorem 3.4 to obtain the following lower bound on the Maximin loss:

**Theorem 4.1.** *There exists a Maximin-optimal $p^*$ such that, for all uniform lotteries $\bar{p}$,*

$$\mathrm{Maximin}(p^*) - \mathrm{Maximin}(\bar{p}) = \Omega\left(\frac{\sqrt{k}}{m}\right).$$

## 4.2 Nash Welfare

As NW has also garnered interest by practitioners and is applicable in practice [18], we upper-bound the NW fairness loss. Unlike MAXIMIN loss, an upper bound on NW loss does not immediately follow from one on $\|\pi - \bar{\pi}\|_\infty$, because decreases in smaller marginals have larger negative impact on the NW. As a result, the upper bound on NW resulting from Section 3 is slightly weaker than that on MAXIMIN:

**Theorem 4.2.** *For* NW*-optimal $p^*$, there exists a uniform lottery $\bar{p}$ that satisfies*

$$\text{NW}(p^*) - \text{NW}(\bar{p}) = \frac{k}{m} \cdot O\left(\min\left\{\sqrt{|\mathcal{C}|\log|\mathcal{C}|}, \ \frac{k}{n_{min}} + 1\right\}\right).$$

We give an overview of the proof of Theorem 4.2. To begin, fix a NW-optimizing panel distribution $p^*, \pi^*$. Before applying our upper bounds on marginal discrepancy from Section 3, we must contend with the fact that if this bounded loss is suffered by already-tiny marginals, the NW may decrease substantially or even go to $0$. Thus, we first prove Lemmas 4.1 and 4.2, which together imply that no marginal in $\pi^*$ is smaller than $1/n$.

**Lemma 4.1.** *For* NW*-optimal $p^*$ over a support of panels $supp(p^*)$, there exists a constant $\lambda \in \mathbb{R}^+$ such that, for all $P \in supp(p^*)$, $\sum_{i \in P} 1/\pi_i^* = \lambda$.*

**Lemma 4.2.** *For* NW*-optimal $p^*, \pi^*$, we have that $\pi_i^* \geq 1/n$ for all $i \in N$.*

Lemma 4.1 follows from the fact that the partial derivative of NW with respect to the probability it assigns a given panel must be the same as that with respect to any other panel at $p^*$ (otherwise, mass in the distribution could be shifted to increase the NW). Lemma 4.2 then follows by the additional observation that $\mathbb{E}_{P \sim p^*}\left[\sum_{i \in P} 1/\pi_i^*\right] = n$.

Finally Lemma 4.3 follows from the fact that Lemma 4.2 limits the potential multiplicative, and therefore additive, impact on the NW of decreasing any marginal by $\|\pi - \bar{\pi}\|_\infty$:

**Lemma 4.3.** *For* NW*-optimal $p^*, \pi^*$, there exists a uniform lottery $\bar{p}, \bar{\pi}$ that satisfies* $\text{NW}(p^*) - \text{NW}(\bar{p}) \leq k \|\pi^* - \bar{\pi}\|_\infty$.

As the NW-optimal marginals $\pi^*$ are anonymous, we can apply the upper bounds given by Theorem 3.3 and Theorem B.8 to show the existence of a $\bar{p}, \bar{\pi}$ satisfying the claim of the theorem.

## 5  Practical Algorithms for Computing Fair Uniform Lotteries

**Algorithms.** First, we implement versions of two existing rounding algorithms, which are implicit in our worst-case upper bounds.[6] The first is Pipage rounding [16], or PIPAGE, a randomized rounding scheme satisfying negative association [10]. The second is BECK-FIALA, the dependent rounding scheme used in the proof of Theorem 3.2. To benchmark these algorithms against the highest level of fairness they could possibly achieve, we use integer programming (IP) to compute the fairest possible uniform lotteries over $supp(p^*)$, the panels over which $p^*$ randomizes.[7] We define IP-MAXIMIN and IP-NW to find uniform lotteries over $supp(p^*)$ maximizing MAXIMIN and NW, respectively. We remark that the performance of these IPs is still subject to our theoretical upper and lower bounds. We provide implementation details in Appendix D.1.

One question is whether we should prefer the IPs or the rounding algorithms for real-world applications. Although IP-MAXIMIN appears to find good solutions at practicable speeds, IP-NW converges to optimality prohibitively slowly in some instances (see Appendix D.2 for runtimes). At the same time, we find that our simpler rounding algorithms give near-optimal uniform lotteries with respect to both fairness objectives. Also in favor of simpler rounding algorithms, many randomized rounding procedures (including Pipage rounding) have the advantage that they exactly

---

[6]We do not implement the algorithm implicit in Theorem 3.3 because our results already present sufficient alternatives for finding excellent uniform lotteries in practice.

[7]Note that these lotteries are not necessarily universally optimal, as they can randomize over only $supp(p^*)$; conceivably, one could find a fairer uniform lottery by also randomizing over panels not in $supp(p^*)$. However, PIPAGE and BECK-FIALA are also restricted in this way, and thus must be weakly dominated by the IP.

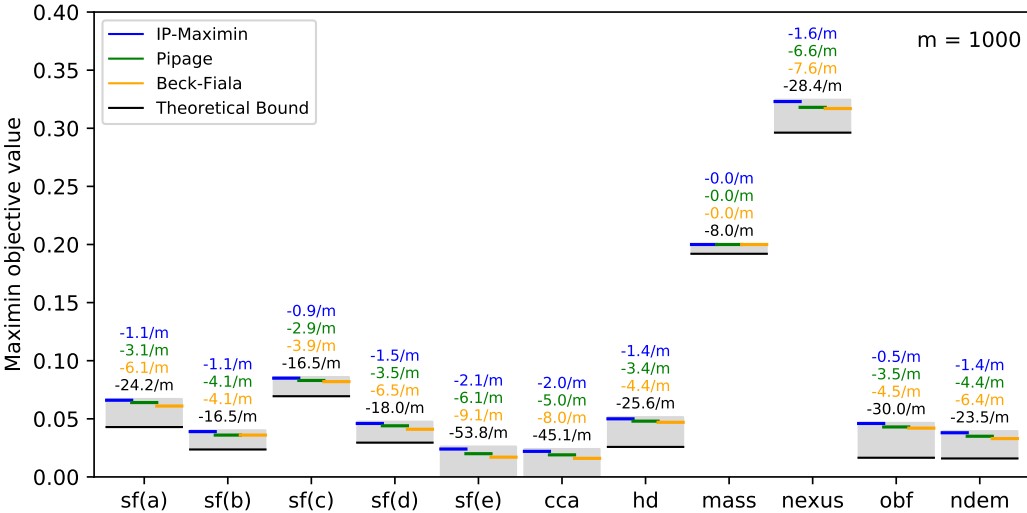

Figure 2: $m = 1000$. Shaded regions extend from $\mathrm{Maximin}(p^*)$, the fairness of the optimal unconstrained distribution, down to the minimum fairness implied by the tightest theoretical upper bound in that instance (in all instances but "obf" Theorem 3.3 is tightest). Each algorithm or bound's loss relative to $\mathrm{Maximin}(p^*)$ is written above in the corresponding color. We show a representative run of PIPAGE, a randomized algorithm.

preserve marginals over the combined steps of randomly rounding to a uniform lottery and then randomly sampling it—a guarantee that is much more challenging to achieve with IPs.

**Uniform lotteries nearly exactly preserve Maximin, Nash Welfare fairness.** We first measure the fairness of uniform lotteries produced by these algorithms in 11 real-world panel selection instances from 7 different organizations worldwide (instance details in Appendix A). In all experiments, we generate a lottery of size $m = 1000$. This is fairly small; it requires drawing only 3 balls from lottery machines, and in one instance we have that $m < n$. We nevertheless see excellent performance of all algorithms, and note that this performance will only improve with larger $m$.

Figure 2 shows the $\mathrm{Maximin}$ fairness of the uniform lottery computed by PIPAGE, BECK-FIALA, and IP-MAXIMIN for each instance. For intuition, recall that the level of $\mathrm{Maximin}$ fairness given by any lottery is exactly the minimum marginal assigned to any individual by that lottery. The upper edges of the gray boxes in Fig. 2 correspond to the optimal fairness attained by an unconstrained distribution $p^*$. These experiments reveal that the cost of transparency to Maximin-fairness is practically non-existent: across instances, the quantized distributions computed by IP-MAXIMIN decrease the minimum marginal by at most $2.1/m$, amounting to a loss of no more than $0.0021$ in the minimum marginal probability in any instance. Visually, we can see that this loss is negligible relative to the original magnitude of even the smallest marginals given by $p^*$. Surprisingly, though PIPAGE and BECK-FIALA do not aim to optimize any fairness objective, they achieve only slightly larger losses in Maximin fairness, with PIPAGE outperforming BECK-FIALA. Finally, the heights of the gray boxes indicate that our theoretical bounds are often meaningful in practice, giving lower bounds on Maximin fairness well above zero in nine out of eleven instances. We note these bounds only tighten with larger $m$. We present similarly encouraging results on NW loss in Appendix D.3.

**Uniform lotteries nearly preserve all Leximin marginals.** We still remain one step away from practice: our examination of $\mathrm{Maximin}$ does not address whether uniform lotteries can attain the finer-tuned fairness properties of the Leximin-optimal distributions currently used in practice. Fortunately, our results from Section 3 imply the existence of a quantized $\bar{p}$ that closely approximates *all marginals* given by the Leximin-optimal distribution $p^*, \pi^*$. We evaluate the extent to which PIPAGE and BECK-FIALA preserve these marginals in Fig. 3. They are benchmarked against a new IP, IP-MARGINALS, which computes the uniform lottery over $supp(p^*)$ minimizing $\|\pi^* - \bar{\pi}\|_\infty$.

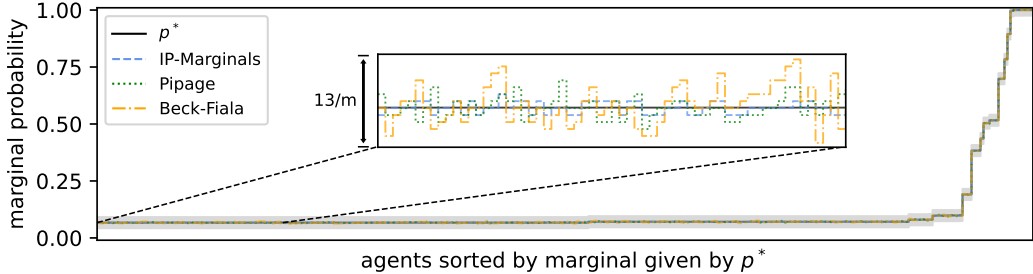

Figure 3: Instance = sf(a), $m = 1000$. Line plot shows the Leximin-optimal marginals $\pi^*$ (implied by panel distribution $p^*$), along with marginals given by all algorithms sorted according to $\pi^*$. Note that each $x$ coordinate then corresponds to an individual. The zoomed box shows the magnitude of marginal discrepancy around $\pi^*$. The surrounding shaded region shows the tightest theoretical bound on the marginal discrepancy, in this case from Theorem 3.3, around the optimal marginals. We show a representative run of PIPAGE, a randomized algorithm.

Figure 3 demonstrates that in the instance "sf(a)", all algorithms produce marginals that deviate negligibly from those given by $\pi^*$. Analogous results on remaining instances appear in Appendix D.4 and show similar results. As was the case for Maximin, we see that our theoretical bounds are meaningful, but that we can consistently outperform them in real-world instances.

## 6  Discussion

Our aim was to show that uniform lotteries can preserve fairness, and our results ultimately suggest this, along with something stronger: that in practical instances, uniform lotteries can reliably almost exactly replicate *the entire set of marginals* given by the optimal unconstrained panel distribution. Our rounding algorithms can thus be plugged directly into the existing panel selection pipeline with essentially no impact on individuals' selection probabilities, thus enabling translation of the output of Panelot (and other maximally fair algorithms) to a nearly maximally fair *and* transparent panel selection procedure. We note that our methods are not just compatible with ball-drawing lotteries, but any form of uniform physical randomness (e.g. dice, wheel-spinning, etc.).

Although we achieve our stated notion of transparency, a limitation of this notion is that it focuses on the final stage of the panel selection process. A more holistic notion of transparency might require that onlookers can verify that the panel is not being intentionally stacked with certain individuals. This work does not fully enable such verification: although onlookers can now observe individuals' marginals, they still cannot verify that these marginals are *actually maximally fair* without verifying the underlying optimization algorithms. In particular, in the common case where quotas require even maximally fair panel distributions to select certain individuals with probability near one, onlookers cannot distinguish those from unfair distributions engineered such that one or more pool members are chosen with probability near one.

In research on economics, fair division, and other areas of AI, randomness is often proposed as a tool to make real-world systems fairer [17, 6, 15]. Nonetheless, in practice, these systems (with a few exceptions, such as school choice [22]) remain stubbornly deterministic. Among the hurdles to bringing the theoretical benefits of randomness into practice is that allocation mechanisms fare best when they can be readily understood, and that randomness can be perceived as undesirable or suspect. Sortition is a rather unique paradigm at the heart of this tension: it relies centrally on randomness, while in the public sphere it is attaining increasing political influence. It is therefore a uniquely high-impact domain in which to study how to combine the benefits of randomness, such as fairness, with transparency. We hope that this work and its potential for impact will inspire the investigation of fairness-transparency tradeoffs in other AI applications.

**Acknowledgements.** We would foremost like to thank Paul Gölz for helpful technical conversations and insights on the practical motivations for this research. We also thank Anupam Gupta for helpful technical conversations. Finally, several organizations for supplying real-world citizens' assembly data, including the Sortition Foundation, the Center for Climate Assemblies, Healthy Democracy, MASS LBP, Nexus Institute, Of by For, and New Democracy.

**Funding and Competing Interests.** This work was partially supported by National Science Foundation grants CCF-2007080, IIS-2024287 and CCF-1733556; and by Office of Naval Research grant N00014-20-1-2488. Bailey Flanigan is supported by the National Science Foundation Graduate Research Fellowship and the Fannie and John Hertz Foundation. None of the authors have competing interests.

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
