# A    Panel Selection Datasets

We examine data from the following 11 real-world sortition panel selection instances, generously provided to us by several groups that specialize in organizing citizens' assemblies. Table 1 shows the instance short-names we use throughout the paper, and which organization was responsible for each panel. The final two columns compare the values of our theoretical upper bounds on the marginal discrepancy, illustrating that in all instances except "obf", the bound from Theorem 3.3 is tighter. Finally, we give some metadata about each instance, which is required for calculating the values of our theoretical upper bounds.

In particular, $n$ = number of pool members, $k$ = number panel members, $\mathcal{C}$ = set of distinct realized feature-vectors in the pool. Precise constants used for computing exact the upper bounds are derived in Appendix B: the Theorem 3.2 bound is exactly $k/m$, the Theorem 3.3 bound is exactly

$$\frac{\sqrt{\frac{1}{2}(1 + \frac{\ln 2}{\ln |\mathcal{C}|})} \cdot \sqrt{|\mathcal{C}| \ln(|\mathcal{C}|)} + 1}{m},$$

and the Theorem B.8 bound is exactly $\frac{2k/n_{min}+1}{m}$. In all instances, $n_{min} = 1$.

Table 1: Instance parameters and resulting theoretical bounds

| Instance | Organization | $n$ | $k$ | $|\mathcal{C}|$ | Thm 3.2 | Thm 3.3 | Thm B.8 |
|---|---|---|---|---|---|---|---|
| sf(a) | Sortition Foundation | 312 | 35 | 182 | $35/m$ | $24.2/m$ | $71/m$ |
| sf(b) | Sortition Foundation | 250 | 20 | 92 | $20/m$ | $16.5/m$ | $41/m$ |
| sf(c) | Sortition Foundation | 161 | 44 | 92 | $44/m$ | $16.5/m$ | $89/m$ |
| sf(d) | Sortition Foundation | 404 | 40 | 108 | $40/m$ | $18.0/m$ | $81/m$ |
| sf(e) | Sortition Foundation | 1727 | 110 | 762 | $110/m$ | $53.8/m$ | $221/m$ |
| cca | Center for Climate Assemblies | 825 | 75 | 554 | $75/m$ | $45.1/m$ | $151/m$ |
| hd | Healthy Democracy | 239 | 30 | 202 | $30/m$ | $25.6/m$ | $61/m$ |
| mass | MASS LBP | 70 | 24 | 25 | $24/m$ | $8.0/m$ | $49/m$ |
| nexus | Nexus | 342 | 170 | 242 | $170/m$ | $28.4/m$ | $341/m$ |
| obf | Of By For | 321 | 30 | 294 | $30/m$ | $31.6/m$ | $61/m$ |
| ndem | New Democracy | 398 | 40 | 173 | $40/m$ | $23.5/m$ | $81/m$ |

# B    Omitted Proofs and Additional Beyond-Worst-Case Upper Bounds from Section 3

## B.1    General Rounding Procedure

Throughout this section, we repeatedly face the task of rounding the entries of some distribution $p$ to some vector $\bar{p}$ that must also be a valid distribution (i.e., have entries in $[0, 1]$ such that $\|\bar{p}\|_1 = 1$), and have entries that are integer multiples of $1/m$. However, many of the standard rounding procedures we apply, such as randomized rounding and discrepancy-based dependent rounding, only give guarantees for rounding probabilities to 0/1 vectors, rather than to multiples of $1/m$. Thus, in several proofs (Theorem 3.1, Theorem 3.2, Theorem 3.3, Theorem B.8), we apply these canonical rounding methods to a *modified version* of our original vector $p$, called $x'$. After constructing $x'$, we round it to a 0/1 vector $\bar{x}'$, from which we finally compute $\bar{p}$. We more precisely define this general rounding procedure, and characterize some of its useful properties, below.

**Definition B.1** (Procedure for using 0/1 rounding procedure to round $p$ to $\bar{p}$). *Let $p$ be a distribution, represented as a vector. Let $x$ be the vector $p$ with entries scaled by $m$, so that $x_j := m \cdot p_j$. Then, define the vector $\lfloor x \rfloor$, which we can think of as the "integer components" of each entry of $x$, i.e., $\lfloor x \rfloor_j := \lfloor m \cdot p_j \rfloor$. Finally, we define $x'$ as the "decimal components" of the entries of $x$, so that $x' := x - \lfloor x \rfloor$. We will round $x'$ to a 0/1 vector.*

*Then, construct $\bar{p}$ from $p$ as follows:*
*1. Construct the vector $x'$ as above.*
*2. Round $x'$ to some 0/1 vector $\bar{x}'$ via a given rounding procedure such that $\|\bar{x}\|_1 = \|x'\|_1$.*

*3. Set $\bar{p}$ such that*

$$\bar{p} := \frac{\lfloor x \rfloor + \bar{x}'}{m}.$$

At a high level, this rounding procedure can be thought of as scaling up the vector we want to round by $m$, holding this scaled vector's integer components aside and rounding its decimal components, and then adding the integer components back in and scaling back down by $m$.

Now, we show that this rounding procedure produces a $\bar{p}$ with the properties we want—(a) it has entries that are multiples of $1/m$ and (b) it is a valid distribution—as well as an additional property (c), which helps translate guarantees on existing roundinig schemes to guarantees in our setting.

**Lemma B.2.** *Suppose we are given a $0/1$ rounding scheme which, given $x' \in [0,1]^{|\mathcal{K}|}$ and constraint matrix $M$, produces some $\bar{x}'$ which satisfies*

- *$\bar{x}' \in \{0,1\}^{|\mathcal{K}|}$,*

- *$\|\bar{x}'\|_1 = \|x'\|_1$, and*

- *$|(M(x' - \bar{x}'))_i| \leq g(i)$ for each row $i$.*

*Then given some distribution $p \in \mathbb{R}_+^{|\mathcal{K}|}$ and $m \in \mathbb{N}$, the procedure in Definition B.1, using such a $0/1$ rounding scheme, produces $\bar{p}$ such that*

*(a) $\bar{p} \in (\mathbb{Z}_+/m)^{|\mathcal{K}|}$,*

*(b) $\bar{p}$ is a distribution, and*

*(c) $|(M(p - \bar{p}))_i| \leq \frac{g(i)}{m}$ for each row $i$.*

*Proof.* We prove each property separately:

(a) holds: $\bar{p}$ contains multiples of $1/m$, since in the general procedure (Definition B.1), its entries are set to the sum of two integers divided by $m$.

(b) holds: $\bar{p}$ is a valid distribution: all entries of $\bar{p}$ must be non-negative, and we have that $\|\bar{p}\|_1 = \|p\|_1 = 1$, as shown below.

$$\|\bar{p}\|_1 = \left\| \frac{\lfloor x \rfloor + \bar{x}'}{m} \right\|_1 = \left\| \frac{\lfloor x \rfloor}{m} \right\|_1 + \left\| \frac{\bar{x}'}{m} \right\|_1 = \left\| \frac{\lfloor x \rfloor}{m} \right\|_1 + \left\| \frac{x'}{m} \right\|_1 = \|p\|_1$$

(c) holds: Fix some $i$ and the corresponding row of $(M(p - \bar{p}))$, referred to as $(M(p - \bar{p}))_i$. Then,

$$|(M(p - \bar{p}))_i| = \left| \left( M \left( \frac{\lfloor x \rfloor + x'}{m} - \frac{\lfloor x \rfloor + \bar{x}'}{m} \right) \right)_i \right| = \frac{|(M(x' - \bar{x}'))_i|}{m} \leq \frac{g(i)}{m}$$

$\square$

## B.2 Omitted Proofs

We will make repeated use of the following generalization of Hoeffding's inequality (see e.g. Proposition 5 of [9]):

**Lemma B.3.** *If $\{\xi_j\}$ are negatively associated random variables with $\xi_j \in [a_j, b_j]$ and $\xi = \sum_j \xi_j$, then*

$$\Pr\left[|\mathbb{E}[\xi] - \xi| \geq t\right] \leq 2\exp\left\{ -\frac{2t^2}{\sum_j (b_j - a_j)^2} \right\}.$$

Here is our first use:

**Theorem 3.1.** *For any realizable $\pi$, we may efficiently randomly generate $\bar{p}$ such that its marginals $\bar{\pi}$ satisfy*

$$\|\pi - \bar{\pi}\|_\infty = O\left( \frac{\sqrt{n \log n}}{m} \right).$$

*Proof of Theorem 3.1.* Given a vector of marginals $\pi$, let $p$ be a basic solution to $Mp = \pi$, where $M$ is the individual-feasible panel membership matrix, so that $|supp(p)| \leq n$.

Then, we will construct $\bar{p}$ from $p$ by constructing $x'$, rounding it to $\bar{x}' \in \{0,1\}^{|\mathcal{K}|}$, and then reconstructing $\bar{p}$ as described in Definition B.1. To do this 0/1 rounding, here we use any randomized rounding procedure that satisfies the following properties: preservation of adding up constraint $\|\bar{x}'\|_1 = \|x'\|_1$, preservation of marginals $E[\bar{x}'_j] = x'_j$, and that $\bar{x}'_j$ are *negatively associated*, as defined in [5, 9]. These properties are satisfied via any number of randomized rounding algorithms [5]. Note as in Definition B.1, $\|\bar{x}'\|_1 = \|x'\|_1$ implies that $\bar{p} \in \bar{\mathcal{D}}$.

Now it remains to analyze the marginal $\bar{\pi}_i$ provided to any given individual $i$ by $\bar{p}$. Consider the collection of $\bar{x}'_j$ for which $i$ is contained in panel $j$. Then, using the negative association of these $\bar{x}'_j$s, we have that for any $t \geq 0$,

$$\Pr[|Mx' - M\bar{x}'| \geq t] = \Pr\left[\left|\mathbb{E}\left[\sum_{j \ni i} \bar{x}'_j\right] - \sum_{j \ni i} \bar{x}'_j\right| \geq t\right], \tag{B.1}$$

by the definition of $\bar{x}'_j$. Then by Hoeffding (Lemma B.3),

$$\leq 2\exp\left(\frac{-2t^2}{|\{j : i \in j\}|}\right) \tag{B.2}$$

$$\leq 2\exp\left(\frac{-2t^2}{n}\right), \tag{B.3}$$

where here we use that $|supp(p)| \leq n$. Then taking $t = \sqrt{\frac{1+\epsilon}{2} n \log n}$,

$$\leq \frac{2}{n^{1+\epsilon}}. \tag{B.4}$$

Taking a union bound over all $n$ rows $i$ then gives

$$\Pr\left[\|Mx' - M\bar{x}'\|_\infty \geq \sqrt{\frac{(1+\epsilon)}{2}} \cdot \sqrt{n \log n}\right] \leq \frac{2}{n^\epsilon} < 1.$$

By Lemma B.2, we therefore have

$$\Pr\left[\|\pi - \bar{\pi}\|_\infty \leq \sqrt{\frac{1+\epsilon}{2}} \cdot \frac{\sqrt{n \log n}}{m}\right] \geq 1 - \frac{2}{n^\epsilon} > 0. \qquad \square$$

Note: if we are additionally guaranteed that all of the $\pi_i = \Omega(k/n)$, then a multiplicative form of Chernoff yields

$$\|\pi - \bar{\pi}\|_\infty = O\left(\sqrt{\frac{k \log n}{mn}}\right)$$

with constant probability.

**Theorem 3.2.** *For any realizable $\pi$, we may efficiently construct $\bar{p}$ such that its marginals $\bar{\pi}$ satisfy*

$$\|\pi - \bar{\pi}\|_\infty \leq k/m.$$

*Proof of Theorem 3.2.* Here, we apply the rounding algorithm used by Flanigan et al. [13] (Lemma 9, Appendix B.4.1), which builds on a notable theorem by Beck and Fiala [1]. Since this rounding algorithm does 0/1 rounding, we apply their algorithm to round $x'$, as in Definition B.1, to some 0/1 vector $\bar{x}'$, from which we construct $\bar{p}$. By Lemma 9 in Appendix B.4.1 in [13], this algorithm ensures the preservation of the "adding up" constraint, that is, that $\|\bar{x}'\|_1 = \|x'\|_1$. Thus, by results (a) and (b) of Lemma B.2, $\bar{p} \in \bar{D}$.

Now, it remains to show that $\|\pi - \bar{\pi}\|_\infty = \|M(p - \bar{p})\|_\infty \leq k/m$. Fortunately, as they prove, the rounding procedure of Flanigan et al. [13] guarantees that when rounding $x'$ to $\bar{x}'$, for a constraint matrix $M$ with column sparsity $k$, $\|M(x' - \bar{x}')\|_\infty \leq k$. By Lemma B.2 result (c), this immediately implies that $\|\pi - \bar{\pi}\|_\infty \leq k/m$. $\qquad \square$

**Theorem 3.3.** *If $\pi$ is anonymous and realizable, then we may efficiently construct $\bar{p}$ such that its marginals $\bar{\pi}$ satisfy*

$$\|\pi - \bar{\pi}\|_\infty = O\left(\frac{\sqrt{|\mathcal{C}|\log|\mathcal{C}|}}{m}\right).$$

*Proof of Theorem 3.3.* We begin with anonymous marginals $\pi$ witnessed by some distribution $p$ over $\mathcal{K}$. The first order of business is to project $p$ into "type space," in order to derive a distribution over panel types. Overloading $F$, we let $F(P) = \mathfrak{P}$ denote the panel type of a given panel $P$, defined as the multiset $F(P) = \{F(i) : i \in P\}$. Then we define the distribution over panel types induced by $p$ as $\mathfrak{p}$, where the probability of drawing panel type $\mathfrak{P}$ from $\mathfrak{p}$ is defined as $\mathfrak{p}_{\mathfrak{P}} := \sum_{P \in \mathcal{K}: F(P) = \mathfrak{P}} p_P$.

This $\mathfrak{p}$ satisfies the PANEL TYPE LP in Eq. (3.3). As an aside, note that this $\mathfrak{p}$ has support $supp(\mathfrak{p}) = \{F(P) : P \in supp(p)\}$. We will assume without loss of generality that $\mathfrak{p}$ is a basic solution to (3.3), so that it has at most $|\mathcal{C}|$ nonzero entries, where $\mathcal{C}$ is the set of all feature-vectors appearing in the pool, i.e., $supp(p) \leq |\mathcal{C}|$. Since $|supp(p)| \leq n$ without loss of generality, $|supp(\mathfrak{p})| \leq n$ also, and so this basic $\mathfrak{p}$ may be found efficiently.

Given this distribution $\mathfrak{p}$ over panel types, we will round it to a uniform lottery $\bar{\mathfrak{p}}$ of size $m$ over panel types $\mathfrak{K}$. Finally, we will lift this distribution over panel types $\bar{\mathfrak{p}}$ back to a distribution $\bar{p}$ over panels with the desired guarantee, and argue that this lift can be performed when the original marginals $\pi$ are anonymous.

We generate $\bar{\mathfrak{p}}$, a distribution with all probabilities multiples of $1/m$, from $\mathfrak{p}$ via randomized rounding, as in Theorem 3.1. To produce $\bar{\mathfrak{p}}$ via a 0/1 rounding algorithm, we follow the procedure given in Definition B.1, where here, $\mathfrak{p}, \bar{\mathfrak{p}}$ correspond to the $p, \bar{p}$ given in the definition. Via this definition, we construct $x, \lfloor x \rfloor, x', \bar{x}'$ analogously, so that $x = m\mathfrak{p}$, etc. By choosing a randomized rounding procedure that preserves $\|\bar{x}'\|_1 = \|x'\|_1$, by Lemma B.2 we have that $\bar{\mathfrak{p}}$ is a valid distribution containing multiples of $1/m$. We again assume this rounding procedure samples $\bar{x}'_j$ which are negatively associated, and preserves that $\mathbb{E}[\bar{x}'_j] = x'_j$ for all panel types $j$.

Recall that type marginals $\tau_c, \bar{\tau}_c$ represent the expected number of panel spots allocated to each feature vector $c$ by $\mathfrak{p}, \bar{\mathfrak{p}}$, respectively, and are given by $\tau = Q\mathfrak{p}$ and $\bar{\tau} = Q\bar{\mathfrak{p}}$. (Recall that $Q$, as described in Section 3, encodes the number of copies of each feature vector on each panel type.) We will next analyze the proximity of the rounded type marginals $\bar{\tau}_c$ to the original type marginals $\tau_c$.

Proceeding via an analysis similar to that of Theorem 3.1, we consider the collection of random variables $\bar{x}'_j$ for which feature vector $c$ appears on panel type $j$ (i.e., $Q_{cj} > 0$). We note that these $\bar{x}'_j$ are again negatively associated, and thus all $Q_{cj}\bar{x}'_j$ are negatively associated, since for a fixed instance all $Q_{cj}$ are constant.

Then for any $t \geq 0$,

$$\Pr[|(Qx' - Q\bar{x}')_c| \geq t] = \Pr\left[\left|\mathbb{E}\left[\sum_j Q_{cj}\bar{x}'_j\right] - \sum_j Q_{cj}\bar{x}'_j\right| \geq t\right], \tag{B.5}$$

by the definition of $x_j$ and $\tilde{x}_j$. Then by Hoeffding (Lemma B.3) with $\xi_j = Q_{cj}\tilde{x}_j$,

$$\leq 2\exp\left(\frac{-2t^2}{\sum_j Q_{cj}^2}\right) \tag{B.6}$$

$$\leq 2\exp\left(\frac{-2t^2}{|\mathcal{C}|m_c^2}\right), \tag{B.7}$$

where $m_c := \max_j Q_{cj}$, and (B.7) uses that for all $c$, $\sum_j Q_{cj}^2 \leq \sum_j m_c^2 \leq |supp(\mathfrak{p})|m_c^2 \leq |\mathcal{C}|m_c^2$. Thus, taking $t_c = \alpha \cdot m_c \cdot \sqrt{|\mathcal{C}|\log|\mathcal{C}|}$,

$$\leq \frac{2}{|\mathcal{C}|^{2\alpha^2}}. \tag{B.8}$$

Taking $\alpha > \sqrt{\frac{1}{2}\left(1 + \frac{\log 2}{\log|\mathcal{C}|}\right)}$ and union bounding over all $|C|$ feature vectors, we may therefore guarantee that with positive probability,

$$|(Qx' - Q\bar{x}')_c| \leq \alpha \cdot m_c\sqrt{|\mathcal{C}|\log|\mathcal{C}|}$$

for all $c$ simultaneously. By Lemma B.2, the derived $\bar{\mathfrak{p}}$ and $\bar{\tau}$ and therefore satisfy

$$|\tau_c - \bar{\tau}_c| \leq \alpha \cdot m_c \frac{\sqrt{|\mathcal{C}| \log |\mathcal{C}|}}{m} \tag{B.9}$$

for all $c$ simultaneously.

Given such a $\bar{\mathfrak{p}}, \bar{\tau}$ over panel types, it remains to construct some uniform lottery $\bar{p}, \bar{\pi}$ over the panels in $\mathcal{K}$ which is consistent with $\bar{\tau}$ and satisfies the desired guarantees on $\bar{\pi}$, which are:

1. each individual appears on each panel in $\bar{p}$ at most once,[8]

2. $0 \leq \bar{\pi}_i \leq 1$ for all $i$, and

3. $|\pi_i - \bar{\pi}_i|$ is small for all $i$.

We will describe a procedure for forming $\bar{p}$ and $supp(\bar{p})$ from $\bar{\mathfrak{p}}$, and then argue that it satisfies all three of these criteria, as well as implies a valid distribution $\bar{p}$ for which all probabilities are multiples of $1/m$. At a high level, this algorithm starts with the panel types $\mathfrak{P}_j$ which form the support of $\mathfrak{p}$, and for each $c$ in turn allocates spots in these panel types $\mathfrak{P}_j$ with feature vector $c$ to individuals in $N_c := \{i \in [n] : F(i) = c\}$, the $n_c$ individuals with feature vector $c$. Given the type marginals $\bar{\tau} = Q\bar{\mathfrak{p}}$ output by our rounding procedure, it first calculates the "ideal" number of spots $\bar{s}_i$ to allocate to each individual $i \in N_c$ across all of $\bar{p}$. It then performs the allocation in such a way that the guarantees above are satisfied. Since $\bar{\mathfrak{p}} \in (\mathbb{Z}_+/m)^{|\mathfrak{K}|}$ and this algorithm populates each $\mathfrak{P}_j$ in the support to create some $P_j \in \mathcal{K}$, it follows that the $\bar{p}$ which it ultimately produces is $\bar{p} \in (\mathbb{Z}_+/m)^{|\mathcal{K}|}$ also.

---

**Algorithm 1** PANELPACKER

---

**Require:** $\bar{\mathfrak{p}} \in (\mathbb{Z}_+/m)^{|\mathfrak{K}|}$ a distribution over feasible panel types, $N$
**Ensure:** $\bar{p} \in (\mathbb{Z}_+/m)^{|\mathcal{K}|}$ a distribution over feasible panels
 1: Initialize $P_j \leftarrow \emptyset$ for each $\mathfrak{P}_j \in supp(\bar{\mathfrak{p}})$, with multiplicity (i.e. for $j \in [m]$)
 2: **for** $c \in \mathcal{C}$ **do**
 3:     Initialize spots $\bar{s}_i \in \{\lfloor m \cdot \bar{\tau}_c/n_c \rfloor, \lceil m \cdot \bar{\tau}_c/n_c \rceil\}$ for $i \in N_c$ such that $\sum_{i \in N_c} \bar{s}_i = m \cdot \bar{\tau}_c$
 4:     Initialize $d_i^1 \leftarrow \bar{s}_i$ for $i \in N_c$
 5:     **for** $j \in [m]$ **do**
 6:         Let $I_{cj}$ be the first $Q_{cj}$ many $i \in N_c$ with largest $d_i^j$
 7:         Update $P_j \leftarrow P_j \cup I_{jc}$
 8:         Update $d_i^{j+1} \leftarrow d_i^j - \mathbb{1}\{i \in I_{cj}\}$ for all $i \in N_c$
 9:     **end for**
10: **end for**
11: **return** $\bar{p}$ the uniform distribution over $P_j$

---

For each panel type $\mathfrak{P}_j$ in the support of $\bar{\mathfrak{p}}$, Algorithm 1 forms one panel in the support of $\bar{p}$ by, for each $c \in \mathcal{C}$, allocating each of panel type $\mathfrak{P}_j$'s $Q_{cj}$ "spots" to individuals $i \in N_c$. It populates each panel type $\mathfrak{P}_j$ with individuals for each $c$ independently. If Algorithm 1 succeeds at step (6) for all $c \in \mathcal{C}$, then it produces a panel $P_j \in supp(\bar{p})$. We first argue that Algorithm 1 succeeds in producing feasible panels.

*Proof that Algorithm 1 succeeds.* In particular, we will argue that Algorithm 1 succeeds for every iteration of step (6). Since $\sum_{i \in N_c} \bar{s}_i = \sum_{\mathfrak{P}_j \in \bar{\mathfrak{p}}} Q_{cj}$, this is equivalent to showing that it assigns all individuals $i \in N_c$ such that $d_i^{m+1} = 0$ for all $i$ and no individual appears on any panel more than once.

---

[8]We note that this is a concern because we will not simply be choosing known panels from collection $\mathcal{K}$, as we don't see the entire collection *a priori*; we will instead be *constructing* panels that must turn out to be feasible.

In each round we have

$$d_i^j := m \cdot \bar{\pi}_i - \sum_{j' < j} \mathbb{1}\{i \in P_{j'}\}$$

the number of spots in $\bar{\mathfrak{p}}$ of type $c$ on which $i$ still needs to be placed at the beginning of round $j$ in order to reach their allocation of $\bar{s}_i$ spots. (This $d_i^j$ can be viewed as the "unsatisfied demand" of individual $i$ at round $j$, according to the promised number of spots $m\bar{\pi}_i$.)

Because the $\bar{\pi}_i$ are all either $\frac{\lfloor m \cdot \bar{\tau}_c / n_c \rfloor}{m}$ or $\frac{\lceil m \cdot \bar{\tau}_c / n_c \rceil}{m}$, the initial values of $d_i^0$ for $i \in N_c$ are all within 1 of one another. Note that step (6) preserves this property that $d_i^j$ remain within 1 of one another for all rounds, since at each step $j$ it decreases some collection of maximal $d_i^j$ by 1.

Suppose for the sake of contradiction that for some $c$, Algorithm 1 reaches some first step $j$ for which a $c$ position on panel $P_j$ cannot be allocated to any $i \in N_c$; then there are not enough individuals with remaining "unmet demand", so $Q_{cj} > |\{i : d_i^j > 0\}|$. Since $Q_{cj} \leq m_c \leq n_c$, it must be the case that some $i \in N_c$ have already been fully assigned by this step $j$ (meaning that for these $i$ it is the case that $d_i^j = 0$), and so all $d_i^j \in \{0, 1\}$ because the $d_i^j$ are within 1 of one another. But $\sum_j Q_{cj} = \sum_i d_i^0 = m \cdot \bar{\tau}_c$, while at this point

$$\sum_{j' \geq j} Q_{cj'} \geq Q_{cj} > |\{i : d_i^j > 0\}| = \sum_i d_i^j,$$

meaning that the number of unallocated positions of type $c$ remaining at step $j$ exceeds the remaining unmet demand of the $i \in N_c$. This implies that strictly more than $Q_{cj'}$ individuals $i$ were given spots on panel $j'$ at step (6) for some earlier $j' < j$. But this is impossible by the definition of Algorithm 1. Therefore Algorithm 1 must succeed in feasibly assigning individuals of each type $c$ to panels.

Since Algorithm 1 succeeds on step (6), it successfully puts $Q_{cj}$ individuals in $N_c$ onto panel $P_j$ for each $j$ and each $c$. By the feasibility of $\mathfrak{P}_j$ we therefore have that $|P_j| = k$ and $P_j$ is quota feasible, since $\mathfrak{P}_j$ is quota feasible and $P_j$ has the exact same numbers of individuals with each feature vector as $\mathfrak{P}_j$.

Therefore Algorithm 1 terminates with a collection of quota-feasible panels, with no individual appearing on any panel more than once. $\square$

We conclude by arguing that the output of Algorithm 1 satisfies the desired guarantees.

First, it is clear that each individual $i$ appears on each panel $P_j \in supp(p)$ at most once. This is because for each individual $i \in N_c$ for some $c$, $i$ is assigned a position on $P_j$ if and only if $i \in I_{cj}$ at step (6), and $I_{cj}$ contains each $i$ at most once by definition. Therefore condition (1) is satisfied.

We next show that these output $\bar{\pi}_i$ satisfy condition (2). For each $i$, its value of $\bar{\pi}_i$ in the distribution $\bar{p}$ output by Algorithm 1 is precisely $\bar{s}_i / m$.

Therefore clearly $\bar{\pi}_i \geq 0$, and since condition (1) holds we have $\sum_j \mathbb{1}\{i \in P_j\} \leq m$, and so $\bar{\pi}_i \leq 1$ also. For a more explicit proof that $\bar{\pi}_i \leq 1$, observe that since $\mathfrak{p}$ is a distribution,

$$\bar{\tau}_c = \sum_j \bar{\mathfrak{p}}_j Q_{cj} \leq \max_j Q_{cj} = m_c \leq n_c,$$

where the last inequality follows because all $\mathfrak{P}$ are feasible panel types, so they cannot contain more individuals $i \in N_c$ than exist in the pool. By Algorithm 1 we have $\bar{s}_i \in \{\lfloor m \cdot \bar{\tau}_c / n_c \rfloor, \lceil m \cdot \bar{\tau}_c / n_c \rceil\}$. Dividing by $n_c$ and multiplying by $m$ yields $\bar{s}_i \leq m$, and so $\bar{\pi}_i = \bar{s}_i / m \leq 1$. Thus (2) is satisfied.

Finally, we confirm condition (3), that the individual marginals are close. By the anonymity of $\pi$, for all $i$ with $F(i) = c$ we have $\pi_i = \tau_c / n_c$, and by its choice of $\bar{s}_i$ and the fact that it succeeds, Algorithm 1 guarantees that $\bar{\pi}_i = \bar{s}_i / m \in (\bar{\tau}_c / n_c - 1/m, \bar{\tau}_c / n_c + 1/m)$. Since $m_c \leq n_c$, therefore (B.9) implies

$$|\pi_i - \bar{\pi}_i| \leq \frac{m_c}{n_c} \cdot \alpha \cdot \frac{\sqrt{|\mathcal{C}| \log |\mathcal{C}|}}{m} + \frac{1}{m} = O\left(\frac{\sqrt{|\mathcal{C}| \log |\mathcal{C}|}}{m}\right),$$

for all $i$, satisfying condition (3) and showing the claim. $\square$

**Theorem 3.4.** *There exist $p, \pi$ for which for all uniform lotteries $\bar{p}, \bar{\pi}$,*

$$\min_{\bar{p} \in \overline{\mathcal{D}}} \|\pi - \bar{\pi}\|_\infty = \Omega\left(\frac{\sqrt{k}}{m}\right).$$

We will make use of the following lemma:

**Lemma B.4.** *Any $k$-uniform hypergraph on $[n]$ is realizable via quotas as the set of feasible panels for an instance of the panel selection problem with pool $[n]$.*

When individual membership in feasible panels is represented as $M \in \{0,1\}^{n \times |\mathcal{K}|}$, this lemma claims that any $M$ with uniform column norms is *realizable* by an instance of the panel selection problem, meaning that there exists an instance of the panel selection problem $(N, k, F, l, u)$ for which $M$ is precisely the individual-panel membership matrix for the set of feasible panels.

*Proof.* Given a set system $\mathcal{S} \subseteq \binom{[n]}{k}$, we may construct a set of upper quotas such that the collection of feasible panels is exactly $\mathcal{S}$.

To do this, construct a binary feature $f_T$ for each $T \notin \mathcal{S}$. For each $i$ in $[n]$, let $f_T(i) = 1$ if and only if $i \in T$; otherwise let $f_T(i) = 0$. Finally, enforce the upper quota that for all feasible panels $P \subset [n]$,

$$\sum_{i \in P} f_T(i) \leq k - 1,$$

for all $T \notin \mathcal{S}$—that is, no feasible panel has more than $k - 1$ members belonging to any $T$. Clearly no $T \notin \mathcal{S}$ is a feasible panel. For $S \in \mathcal{S}$, observe that $|S| = k$, and so for all $T \notin \mathcal{S}$, we have $|S \cap T| \leq k - 1$. Therefore all $S \in \mathcal{S}$ are feasible.

Finally, it bears noting that this is also possible to execute using lower quotas: taking $f'_T(i) = 1 - f_T(i)$, we could instead enforce for each $T \notin \mathcal{S}$ that

$$\sum_{i \in P} f'_T(i) \geq 1.$$

$\square$

*Proof of Theorem 3.4.* Using Lemma B.4, our aim is to identify and deploy some matrix $M \in \{0,1\}^{n \times |\mathcal{K}|}$ for which

$$\min_{\bar{x} \in \bar{\Delta}} \|M\bar{x}\|_\infty = \Omega\left(\sqrt{k}\right),$$

where $\bar{\Delta} := \{x \in \{\ldots, -3, -1, 1, 3, \ldots\}^n : \sum_i x_i = 0\}$ and all columns of $M$ sum to $k$. Translating and scaling appropriately and applying Lemma B.4, this will provide our desired $\Omega\left(\frac{\sqrt{k}}{m}\right)$ lower bound.

The common instances which provide lower bounds of $\Omega(\sqrt{k})$ for the Beck-Fiala problem are insufficient for our purposes in two respects. First, while they are column-sparse, they are generally not uniform in column norm. Second, they are incomparable in terms of the $\bar{x}$ which they quantify over: the Beck-Fiala problem considers minimizing $\|M\bar{x}\|_\infty$ in the more restrictive rounding setting where $\bar{x} \in \{-1, 1\}^n$, while we are concerned with $\bar{x} \in \bar{\Delta}$.

We overcome these barriers by first modifying the Walsh matrices — a family of Hadamard matrices — in order to guarantee uniform column norms, and then modifying the Beck-Fiala lower bound proof of [25, Theorem 19] for arbitrary Hadamard matrices to apply to our matrices for all $\bar{x} \in (2\mathbb{Z} + 1)^n$.

To begin, let $H_t$ be the $2^t \times 2^t$ Walsh matrix, defined recursively by $H_0 = 1$ and

$$H_{t+1} = \begin{bmatrix} H_t & H_t \\ H_t & -H_t \end{bmatrix}.$$

Let $N := 2^t$ denote its dimension.[9] It is a fact that all rows (and columns) besides the first have an equal number of $1$ and $-1$ entries. Therefore we take $H'_t$ to be the submatrix derived by dropping

---

[9]Note that this $N$ is a variable used only in this proof, and it is unrelated to the pool $N$ and its magnitude $n$ as used in the paper body.

the first two columns of $H_t$. (We remove the first column so that all remaining columns have equal sum; we remove the second so that $\bar{\Delta}$ is nonempty). Additionally, let $h_i$ denote the rows of $H'_t$, and $h^j$ denote its columns. Then $H'_t$ has the property that $\sum_i h_i^j = 0$, and in particular all columns $h^j$ have $N/2$ 1-entries.

We have the following lemma:

**Lemma B.5.**
$$\min_{x \in \bar{\Delta}} \|H'_t x\|_\infty \geq \frac{N-2}{\sqrt{N}},$$

*where* $\bar{\Delta} := \{x \in \{\ldots, -3, -1, 1, 3, \ldots\}^{N-2}\}$.

*Proof.* This right-hand side is $H'_t x = (h_1 x, \ldots, h_N x)^T$. We aim to show that there is some $i$ for which $|h_i x|$ is large. Writing $\|H'_t x\|_2^2$ two ways, we have that

$$\sum_i (h_i x)^2 = \|x_1 h^1 + \ldots + x_{N-2} h^{N-2}\|_2^2$$

$$= \sum_j x_j^2 \|h^j\|_2^2 + \sum_{j \neq k} x_j x_k (h^j \cdot h^k).$$

The entries of $H_t$ are all $\pm 1$, and $h^j \cdot h^k = 0$ for $j \neq k$ (since the columns of $H_t$ and therefore $H'_t$ are orthogonal), so this becomes

$$= (N-2) \sum_j x_j^2$$

$$\geq (N-2)^2,$$

since $x_i^2 \geq 1$ by assumption. Therefore by averaging there is some $i$ for which $(h_i x)^2 \geq \frac{(N-2)^2}{N}$, and so $|h_i x| \geq \frac{N-2}{\sqrt{N}})$, as desired. $\square$

Next we translate $H'_t$ into an instance of the panel selection problem and argue it has the desired properties. Take $M := \frac{1}{2}(H_t + 1^{N \times (N-2)})$ to be the $\{0, 1\}$ matrix derived from $H'_t$.

The fact that $M$ has uniform column norm $k = N/2$ directly follows from a property of Walsh matrices. Therefore we may apply Lemma B.4 to argue that $M$ is realizable as the individual-panel membership matrix for some instance of the panel selection problem, with $n = N$, $|\mathcal{K}| = N - 2$, and $k = N/2$.

To conclude, consider the uniform $p = \left(\frac{1}{N-2}, \ldots, \frac{1}{N-2}\right)$, with $m = a(N-2) + (N-2)/2$ for any $a \in \mathbb{Z}_+$. In this case, each coordinate of $p$ falls evenly between multiples of $1/m$ and must be rounded to multiples of $1/m$. Letting $x := p - \lfloor mp \rfloor / m = (1/2m, \ldots, 1/2m)$ be this vector of remainders, we must replace it with some $\bar{x} \in (\mathbb{Z}/m)^{N-2}$, while maintaining that $\sum_j \bar{x}_j = \sum_j x_j = (N-2)/2m$, so that the resulting $\bar{p} = \lfloor mp \rfloor / m + \bar{x}$ remains a distribution over panels. (Note that here negative $\bar{x}_j$ signify that the distribution mass on panel $j$ decreases from $p$ to $\bar{p}$.)

Explicitly, we then have

$$\|\pi - \bar{\pi}\|_\infty = \|Mp - M\bar{p}\|_\infty \tag{B.10}$$

$$= \|M(x - \bar{x})\|_\infty \tag{B.11}$$

$$= \frac{1}{2m}\|My\|_\infty, \tag{B.12}$$

where $y := 2m(\bar{x} - x)$.

$$= \frac{1}{2m}\|\frac{1}{2}H'_t y + \frac{1}{2}1^{N \times (N-2)} y\|_\infty \tag{B.13}$$

$$= \frac{1}{4m}\|H'_t y\|_\infty, \tag{B.14}$$

where $\sum_i y_i = 0$ because we require that $\bar{p}$ remain a distribution. Then since $y \in (2\mathbb{Z}+1)^{N-2}$, by Lemma B.5 we have

$$\geq \frac{N-2}{4m\sqrt{N}} \tag{B.15}$$

$$= \Omega\left(\frac{\sqrt{k}}{m}\right), \tag{B.16}$$

since $k = N/2$.

This holds for all $y \in (2\mathbb{Z}+1)^{N-2}$. Recall that $\overline{\mathcal{D}} := \{\bar{p} \in (\mathbb{Z}_+/m)^{|\mathcal{K}|} : \|\bar{p}\|_1 = 1\}$, and so

$$\overline{\mathcal{D}} \subseteq \{p + \bar{\Delta}/2m\}.$$

Therefore (B.16) implies that

$$\min_{\bar{p} \in \overline{\mathcal{D}}} \|\pi - \bar{\pi}\|_\infty = \Omega\left(\frac{\sqrt{k}}{m}\right),$$

as desired. $\qquad\square$

## B.3 Additional Beyond-Worst-Case Upper Bounds

Since some of our beyond-worst-case upper bounds apply to anonymous realizable $\pi$, it is reasonable to ask how prevalent anonymous realizable $\pi$ are, for arbitrary instances of sortition. Fortunately, we have the following claim:

**Claim B.6.** *For any instance of the panel selection problem and any realizable $\pi$, let $\pi'$ be the "anonymized" marginals obtained by setting $\pi'_i$ to the average $\pi_{i'}$ across all $i'$ with the same feature vector as $i$. Then $\pi'$ is realizable also.*

*Proof of Claim B.6.* Let $\pi^*$ denote the "anonymization" of $\pi$, and take

$$\Pi := \left\{ \pi' : \text{ realizable, and for all } c, \sum_{i:F(i)=c} \pi'_i = \sum_{i:F(i)=c} \pi_i \right\}.$$

We will show that $\pi^* \in \Pi$.

We argue by way of contradiction. Let $\hat{\pi}$ denote the "most anonymized" $\pi' \in \Pi$, in the sense that

$$\hat{\pi} = \arg\min_{\pi' \in \Pi} \max_c \left( \max_{i:F(i)=c} \pi'_i - \min_{i:F(i)=c} \pi'_i \right).$$

Let $i$ and $i'$ be some pair of individuals with $F(i) = F(i')$ witnessing this maximum diameter, and let $p$ be a distribution with marginals $\hat{\pi}$. For each such pair, we will argue that $p$ may be modified so that $\hat{\pi}_i = \hat{\pi}_{i'}$ while leaving all other marginals unchanged. By iteratively applying this to all such pairs, we will contradict the minimality of $\hat{\pi}$.

To start, observe that by assumption $\hat{\pi}_i > \hat{\pi}_{i'}$. Let $p'$ be the distribution over feasible panels which is the same as $p$, except that $i$ and $i'$ switch places in any panel on which either of them appear. All such panel replacements yield feasible panels, since they have the same feature vector $c$. Finally take $p_{new} = (p + p')/2$. As promised, this distribution has the property that $\pi_i = \pi_{i'}$ and all other marginals are unchanged. $\qquad\square$

As a belated warm-up to the beyond-worst-case guarantees, we address the case when there is only one feature of interest, so that $F = \{f\}$. It turns out that we can obtain strong guarantees for this special case without using the machinery deployed in the proof of Theorem 3.3. We place no constraints on the size of the set of feature values $\Omega$, nor do we require that $\pi$ is anonymous.

**Theorem B.7.** *If $\pi$ is realizable and $|F| = 1$, then we may efficiently identify $\bar{p}$ such that its marginals $\bar{\pi}$ satisfy*

$$\|\pi - \bar{\pi}\|_\infty < \frac{2}{m}.$$

*Proof of Theorem B.7.* Given marginals $\pi$, let $p$ be a distribution over feasible panels $\mathcal{K}$ which witnesses $\pi$. The first step of this rounding is to consider the marginals $\tau_v$ of each feature value $v$: $\tau_v = \sum_{i:f(i)=v} \pi_i$. Note that $\sum_v \tau_v = \sum_i \pi_i = k$. Since there is only one feature, all feasible panels $P$ satisfy

$$l_v \le |\{i \in P : f(i) = v\}| \le u_v, \tag{B.17}$$

and taking the expectation of this over $p$ gives

$$l_v \le \mathbb{E}_p[|\{i \in P : f(i) = v\}|] \le u_v \tag{B.18}$$

$$l_v \le \tau_v \le u_v. \tag{B.19}$$

Therefore $l_v \le \lfloor \tau_v \rfloor$ and $u_v \ge \lceil \tau_v \rceil$. We will construct a new distribution $\bar{p}$ over panels $P$ which satisfy $\lfloor \tau_v \rfloor \le |\{i \in P : f(i) = v\}| \le \lceil \tau_v \rceil$ for all features $v$, and are therefore guaranteed to be feasible.

We will construct feasible panels via the following scheme. Consider the interval $[0, km] \subset \mathbb{R}$ as representing the $km$ spots to be allocated across the $m$ panels which will comprise our lottery, and let $s_t := [t-1, t)$ denote spot $t$. Next observe that $m \sum_i \pi_i = km$, and so $m\pi_i$ may be viewed as the expected number of spots which $p$ would give to $i$.

First group the $\pi_i$ by feature value to form $\tau_v = \sum_{i:f(i)=v} \pi_i$, and then pack them into $[0, km]$, so that individuals with common feature values have contiguous sections; let $S_i$ denote the portion of $[0, km]$ allocated to $i$, so that $|S_i| = \pi_i$. We will choose an individual $I(t)$ for each spot $s_t$, and then assemble the $m$ panels that comprise $\bar{p}$ by taking

$$P_r := \{I(t) : t = wm + r \text{ for } w \in \{0, \ldots, k-1\}\}, \tag{B.20}$$

for $r \in \{1, \ldots, m\}$.

How to choose which individual will get the spot $t$ for each $t$? If $S_i \supseteq s_t$ then $I(t) = i$. Otherwise, $s_t$ is split between two or more individuals, possibly with different feature values, in which case we call it *contested*. Observe that no matter how these contested $s_t$ are allocated (no matter the choice of $I(t)$ for split $t$), it will be the case that $|\pi_i - \bar{\pi}_i| < 2/m$, since there is at most one contested $s_t$ at each endpoint of the interval $S_i$.

It remains to argue that the panels chosen in (B.20) are feasible; in particular that $\lfloor \tau_v \rfloor \le \bar{\tau}_v \le \lceil \tau_v \rceil$ for all $v$. By construction, each panel $P_r$ has some number of spots which will necessarily be allocated to an individual with feature value $v$, and some number of spots which are contested and may or may not be allocated to an individual with feature vector $v$. For each value $v$, there are at most two spots in all of $[0, km]$ which are *type contested* in this way. If some panel $P_r$ contains at most one type-contested spot for type $v$, then no matter which way it is allocated, $|\{i \in P : f(i) = v\}| - \tau_v| < 1$, and so $P_r$ is feasible with respect to $v$. In the worst case, for some given $v$ both of the spots which are type-contested by $v$ appear on the same panel $P_r$. In order to ensure that $|\{i \in P : f(i) = v\}| - \tau_v| < 1$, it must be the case that exactly one of these two spots is allocated to some $i$ for which $f(i) = v$. Fortunately this constraint is easily satisfiable, even in the case when a given panel $P_r$ contains both of the type-contested spots for multiple features $v$.

Therefore the $\bar{p}$ as constructed by (B.20) is supported by panels which are not only feasible but respect quotas which are maximally tight, given that the input $p, \pi$ was realizable. Finally since each $i$ contests at most two spots, we have that

$$\|\pi - \bar{\pi}\|_\infty < \frac{2}{m}. \qquad \square$$

**Theorem B.8.** *Given realizable anonymous $\pi$, we may efficiently identify $\bar{p}, \bar{\pi}$ such that*

$$\|\pi - \bar{\pi}\|_\infty = O\left(\frac{1}{m} \max\left\{\frac{k}{n_{min}}, 1\right\}\right),$$

*where $n_{min} := \min_c n_c$ is the minimum number of individuals in the pool which share any one feature vector.*

*Proof.* We proceed as in the proof of Theorem 3.3, but apply a different rounding to the panel type LP to obtain $\bar{p}$. To begin, $p, \pi$ projects to some $\mathfrak{p}, \tau$. Without loss of generality assume that it is a basic solution to the TYPE LP (3.4).

We will construct $\bar{\mathfrak{p}}$ from $\mathfrak{p}$ by applying $0/1$ rounding as in Definition B.1.

Note that the constraint matrix $Q$ in (3.3) has the property that for all columns $q^j$, $\|q^j\|_1 = k$. As a special case of [8, Theorem 6], applied to $x'$ and the panel type LP, there exists an $\bar{x}' \in \{0,1\}^{|\mathfrak{R}|}$ such that

$$\|Q(x' - \bar{x}')\|_\infty < 2k.$$

and for which $\|\bar{x}'\|_1 = \|x\|_1$. (This follows from a generalization of the Beck-Fiala algorithm which both respects hard constraints and applies to arbitary matrices $Q$ with bounded column norms, and is therefore also algorithmic.)

Applying Lemma B.2, we then have

$$\|\tau - \bar{\tau}\|_\infty < \frac{2k}{m}.$$

Given that such a $\bar{\mathfrak{p}}, \bar{\tau}$ exists, it remains to generate $\bar{p}$ and $\bar{\pi}$ in such a way as to give the desired bound on the discrepany in individual marginals. We proceed in a manner identical to the proof of Theorem 3.3.

Again we have that $\bar{\tau} \geq 0$ and $\bar{\tau} = \sum_j Q_{cj}\bar{p}_j \leq m_c \leq n_c$, where $m_c = \max_j Q_{cj}$ and $n_c$ is the number of individuals $i$ for which $F(i) = c$, since $\bar{\mathfrak{p}}$ is a distribution over feasible panel types $j$. Therefore dividing $\bar{\tau}$ amongst the $\bar{\pi}_i$ as equally as possible for each $c$ gives $\bar{\pi}_i \in [0,1]$.

By the anonymity of $\pi$, for all $i$ with $F(i) = c$, $\pi_i = \tau_c/n_c$, and dividing the spots in $\bar{\mathfrak{p}}$ for feature vector $c$ as equally as possible amongst the $n_c$ individuals gives $\bar{\pi}_i \in \{\bar{\tau}_c/n_c \pm \frac{1}{m}\}$. This equal division of spots in order to form $\bar{p}$ from $\bar{\mathfrak{p}}$ is feasible by the same Algorithm 1 as in the proof of Theorem 3.3. Therefore the resulting $\bar{p}, \bar{\pi}$ satisfies

$$\begin{aligned}
\|\pi - \bar{\pi}\|_\infty &= \max_c |\tau_c/n_c - \bar{\pi}| \\
&< \frac{1}{n_c}\|\tau - \bar{\tau}\|_\infty + \frac{1}{m} \\
&< \frac{2k}{n_{min} \cdot m} + \frac{1}{m}. \qquad \Box
\end{aligned}$$

## C   Omitted Proofs from Section 4

**Theorem 4.1.** *There exists a Maximin-optimal $p^*$ such that, for all uniform lotteries $\bar{p}$,*

$$\mathrm{Maximin}(p^*) - \mathrm{Maximin}(\bar{p}) = \Omega\left(\frac{\sqrt{k}}{m}\right).$$

*Proof of Theorem 4.1.* We will follow the proof of Theorem 3.4: first we use the Walsh matrices to construct a matrix with the desired properties, prove a modified version of Lemma B.5 for it, and then appeal to Lemma B.4 to argue that it corresponds to a realizable instance of the panel selection problem.

In contrast to the construction in Theorem 3.4, where we need only demonstrate that *some* $\bar{\pi}_i$ deviates from $\pi_i$, we must construct an instance for which (essentially) the minimum $\pi_i$ necessarily *decreases*. We accomplish this by first modifying the Walsh matrices to have uniform row norm, so that $\pi$ is uniform and all $\pi_i$ are minimal. We then introduce a second set of "twin" individuals, each $i'$ of which is a member of the panels which their twin $i$ is not. This ensures that any discrepancy in $\bar{\pi} - \pi$ is witnessed in the downward direction.

To begin, again let $H_t$ be the $2^t \times 2^t$ Walsh matrix, with $N := 2^t$ its dimension. This time we take $H_t^*$ to be the submatrix derived by dropping the first row of $H_t$. By properties of Walsh matrices, all remaining rows in $H_t^*$ have an equal number of $1$ and $-1$ entries, (though this is no longer true of the columns).

Again letting $h_i$ denote the rows of $H_t^*$, and $h^j$ denote its columns, we have the following new version of Lemma B.5, which requires the additional assumption that $\sum_j x_j = 0$:

**Lemma C.1.**
$$\min_{x \in \Delta^*} \|H_t^* x\|_\infty \geq \sqrt{N},$$
*where* $\Delta^* := \{x \in \{\ldots, -3, -1, 1, 3, \ldots\}^N : \sum_j x_j = 0\}$.

*Proof.* This right-hand side is $H_t' x = (h_1 x, \ldots, h_{N-1} x)^T$. We aim to show that there is some $i$ for which $|h_i x|$ is large. Writing $\|H_t' x\|_2^2$ two ways, we have that

$$\sum_i (h_i x)^2 = \|x_1 h^1 + \ldots + x_N h^N\|_2^2$$

$$= \sum_j x_j^2 \|h^j\|_2^2 + \sum_{j \neq k} x_j x_k (h^j \cdot h^k)$$

the entries of $H_t^*$ are all $\pm 1$, and $h^j \cdot h^k = -1$ for $j \neq k$ (since the columns of $H_t$ were orthogonal), so this becomes

$$= (N-1) \sum_j x_j^2 - \sum_{j \neq k} x_j x_k$$

$$= N \sum_j x_j^2 - \sum_j \sum_k x_j x_k$$

$$= N \sum_j x_j^2$$

$$\geq N^2,$$

since $x_i^2 \geq 1$ by assumption. Therefore by averaging, there is some $i$ for which $(h_i x)^2 \geq \frac{N^2}{N-1}$, and so $|h_i x| \geq \sqrt{N}$, as desired. $\square$

As constructed, all rows of $H_t^*$ have the same number of 1s, so when we transform it into some $M$ for some instance of the panel selection problem, it will yield that the marginals $\pi$ of uniform $p$ are uniform. However we cannot yet apply Lemma B.4, since the columns of the resulting $M$ do not have constant norm; in particular, the first column will be all 1s.

In order to simultaneously correct for this and translate from $\ell_\infty$ to Maximin lower bounds, we introduce "twins" for each $i$. Letting $M^* = \frac{1}{2}(H_t^* + 1^{(N-1) \times N})$ be this $\{0, 1\}$ matrix, define $\bar{M}^* := 1^{(N-1) \times N} - M^*$ to be its complement, so that $M_{ij}^* = 1 - \bar{M}_{ij}^*$ for all $i, j$. Finally take

$$M = \begin{bmatrix} M^* \\ \bar{M}^* \end{bmatrix}$$

and observe that this $M \in \{0, 1\}^{(2N-2) \times N}$ has uniform column norm $N-1$ because of $\bar{M}^*$. We may therefore apply Lemma B.4 to claim that it is the individual-panel membership matrix of some instance of the panel selection problem.

The remainder of the argument proceeds similarly to that of Lemma B.5, with additional step of showing that the lower bound holds for the maximin objective. We include the full argument for completeness.

Similarly take $p = \left(\frac{1}{N}, \ldots, \frac{1}{N}\right)^T$, with $m = aN + N/2$ for any $a \in \mathbb{Z}_+$, $n = 2N - 2$ (the number of individuals), and $k = N - 1$. This $p$ gives equal marginals: here $\pi_i = (Mp)_i = \frac{N-1}{2N-2} = \frac{k}{n}$ for all $i$. Again each coordinate of $p$ falls evenly between multiples of $1/m$ and must be rounded to multiples of $1/m$. Letting $x := p - \lfloor mp \rfloor / m = (1/2m, \ldots, 1/2m)^T$ be this vector of remainders, we must replace it with some $\bar{x} \in (\mathbb{Z}/m)^N$, while maintaining that $\sum_j \bar{x}_j = \sum_j x_j = N/2m$, so that the resulting $\bar{p} = \lfloor mp \rfloor / m + \bar{x}$ remains a distribution over panels.

Explicitly, we then have

$$\|\pi - \bar{\pi}\|_\infty = \|Mp - M\bar{p}\|_\infty \tag{C.1}$$

$$= \|M(x - \bar{x})\|_\infty \tag{C.2}$$

$$= \frac{1}{2m} \left\| \begin{bmatrix} M^* \\ \bar{M}^* \end{bmatrix} y \right\|_\infty, \tag{C.3}$$

where $y := 2m(\bar{x} - x)$. Because $\ell_\infty$ is a maximum, this is

$$\geq \frac{1}{2m} \|M^* y\|_\infty \tag{C.4}$$

$$= \frac{1}{2m} \|\frac{1}{2} H_t^* y + \frac{1}{2} 1^{(N-1) \times N} y\|_\infty \tag{C.5}$$

$$= \frac{1}{4m} \|H_t^* y\|_\infty, \tag{C.6}$$

where $\sum_i y_i = 0$ because we require that $\bar{p}$ remain a distribution. Then since $y \in (2\mathbb{Z} + 1)^{N-2}$, by Lemma B.5 we have

$$\geq \frac{\sqrt{N}}{4m} \tag{C.7}$$

$$= \Omega\left(\frac{\sqrt{k}}{m}\right), \tag{C.8}$$

since $k = N - 1$. Again since $\overline{\mathcal{D}} \subseteq \{p + \bar{\Delta}/2m\}$, we then have

$$\min_{\bar{p} \in \overline{\mathcal{D}}} \|\pi - \bar{\pi}\|_\infty = \Omega\left(\frac{\sqrt{k}}{m}\right).$$

Since $\pi$ is uniform by construction (and so these $p$ and $\pi$ are optimal with respect ot Maximin), this is a lower bound on the discrepancy of each marginal which was minimal *before deviation*. It finally remains to show that this deviation happens in the downward direction, so that the minimum marginal decreases by at least this amount. Observe that by the construction of $\bar{M}^*$, for all $\bar{p}$ we have $(M^* \bar{p})_i = -(\bar{M}^* \bar{p})_i$. Therefore for any given $\bar{p}$, whichever coordinate $i$ satisfies $|(\pi - \bar{\pi})_i| = \Omega(\sqrt{k}/m)$, there is a coordinate $i'$ for which $(\pi - \bar{\pi})_{i'} = \Omega(\sqrt{k}/m)$. Therefore in this instance

$$\text{Maximin}(p^*) - \max_{\bar{p} \in \overline{\mathcal{D}}} \text{Maximin}(\bar{p}) = \Omega\left(\frac{\sqrt{k}}{m}\right),$$

as desired. $\qquad\square$

**Lemma 4.1.** *For* NW*-optimal $p^*$ over a support of panels $supp(p^*)$, there exists a constant $\lambda \in \mathbb{R}^+$ such that, for all $P \in supp(p^*)$, $\sum_{i \in P} 1/\pi_i^* = \lambda$.*

*Proof of Lemma 4.1.* We can write the problem of finding the NW optimizing distribution over a fixed panel support $\mathcal{P} \subseteq \mathcal{K}$ as below on the left, where $NW^n(p)$ is equal to the product of the $\pi_i$, the marginals implied by the panel distribution $p$ (in contrast, in Section 2, we let $NW(p)$ be the geometric mean—here we we take the $n^{th}$ power). On the right, we've rewritten the program in standard form, where we set $f(p) = -NW^n(p)$, $h(p) = p_1 + p_2 + \cdots + p_{|\mathcal{P}|} - 1$, and $g_j(p) = -p_j$. Observe that, $\forall j \in [|\mathcal{P}|]$, $\nabla h(p) = 1$ and $\nabla g_j(p) = -e_j$, where $e_j$ is the vector of 0s with a 1 at index $j$.

$$\max_p NW^n(p) \qquad\qquad \min_p f(p)$$
$$\|p\|_1 = 1 \qquad\qquad h(p) = 0$$
$$p_j \geq 0 \; \forall j \in [|\mathcal{P}|] \qquad\qquad g_j(p) \leq 0 \; \forall j \in [|\mathcal{P}|]$$

Now, let $p^*$ be an optimal solution to this program, and $supp(p^*)$ be its support, i.e., the set of panels to which $p^*$ assigns nonzero probability. Then, since the objective and constraints of the above program are continuously differentiable over their entire support (and thus at $p^*$), by the KKT condition Stationarity, there exist some constants $\lambda$ and $\mu_j$ for all $j \in [|supp(p^*)|]$ (where $\mathbf{0}$ is the zero vector) such that

$$\nabla f(p^*) + \lambda \nabla h(p^*) + \sum_{j \in [|supp(p^*)|]} \mu_j \nabla g(p^*) = \mathbf{0} \implies (\nabla f(p^*))_j = \mu_j - \lambda$$

By dual feasibility and primal feasibility respectively, we have that $\mu_j, p_j \geq 0$ for all $j \in [|supp(p^*)|]$; by complementary slackness, we have that $\sum_{j \in [|supp(p^*)|]} \mu_j p_j^* = 0$. Thus, for all

$j$, either $p_j^* = 0$, or $p_j^* > 0$ and $\mu_j = 0$. We have restricted $supp(p^*)$ to panels $j$ in which $p_j^* > 0$, so we conclude that $\mu_j = 0$. It follows that

$$\frac{\partial NW^n(p^*)}{\partial p_j^*} = -(\nabla f(p^*))_j = -(\mu_j - \lambda) = \lambda \quad \forall j \in supp(p^*)$$

Finally, we can conclude the proof by expressing this partial derivative for fixed $p_j$ (which as shown, has a constant value across all $j$ in the support) in terms of the marginals $\pi$. We obtain that for all $j$ in $supp(p^*)$,

$$\lambda = \frac{\partial NW^n(p^*)}{\partial p_j^*} = \sum_{i \in N} \frac{NW^n(p^*)}{\pi_i^*} \frac{\partial \pi_i^*}{\partial p_j^*} = \sum_{i \in P_j} \frac{NW^n(p^*)}{\pi_i^*} = NW^n(p^*) \left( \sum_{i \in P_j} \frac{1}{\pi_i^*} \right)$$

where $P_j$ is the $j^{th}$ panel in $supp(p^*)$. The second equality is by the product rule for derivatives, where each term of the resulting sum is equal to the derivative of $\pi_i^*$ with respect to $p_j^*$ multiplied by $NW/\pi_i^*$, the NW holding out the marginal of individual $i$. The third equality is by the fact that if $i \in P_j$, then $\partial \pi_i^*/\partial p_j^* = 1$; otherwise $\partial \pi_i^*/\partial p_j^* = 0$. $\qquad\square$

**Lemma 4.2.** *For* NW-*optimal* $p^*, \pi^*$, *we have that* $\pi_i^* \geq 1/n$ *for all* $i \in N$.

*Proof of Lemma 4.2.* Let $X[P \ni i]$ be the indicator that a panel $P$ contains individual $i$. Then,

$$\mathbb{E}_{P \sim p^*} \left[ \sum_{i \in P} \frac{1}{\pi_i^*} \right] = \mathbb{E}_{P \sim p^*} \left[ \sum_{i \in N} \frac{X[P \ni i]}{\pi_i^*} \right] = \sum_{i \in N} \frac{\mathbb{E}_{P \sim p^*}[X[P \ni i]]}{\pi_i^*} = \sum_{i \in N} \frac{\pi_i^*}{\pi_i^*} = n$$

By Lemma 4.1, we also have that $\mathbb{E}\left[ \sum_{i \in P} \frac{1}{\pi_i^*} \right] = \lambda/NW^n(p^*)$, and thus $\lambda/NW^n(p^*) = n$. It follows that for all panels $P$, $\sum_{i \in P} \frac{1}{\pi_i^*} = \lambda/NW^n(p^*) = n$ and therefore $\pi_i^* \geq 1/n \; \forall i \in N$; otherwise, we would have some panel $P$ for which $\sum_{i \in P} \frac{1}{\pi_i} > n$, a contradiction. $\qquad\square$

**Lemma 4.3.** *For* NW-*optimal* $p^*, \pi^*$, *there exists a uniform lottery* $\bar{p}, \bar{\pi}$ *that satisfies* $NW(p^*) - NW(\bar{p}) \leq k \|\pi^* - \bar{\pi}\|_\infty$.

*Proof of Lemma 4.3.* Let $\pi_{min}^*$ be the smallest marginal of any individual implied by the Nash-optimal distribution over panels $p^*$, i.e., $\pi_{min}^* = \min_{i \in N} \pi_i^*$. Then, to upper-bound the loss in NW, we assume an unattainable worst case that between $p^*, \pi^*$ and a given uniform lottery $\bar{p}, \bar{\pi}$, *all* individuals probabilities suffer the largest loss of any marginal, $\|\pi^* - \bar{\pi}\|_\infty$, and that this loss manifests multiplicatively as badly as if all agents had original marginal probability $\pi_{min}^*$. This first gives the multiplicative bound:

$$NW(\bar{p}^*) \geq NW(p^*) \left( \frac{\pi_{min}^* - \|\pi^* - \bar{\pi}\|_\infty}{\pi_{min}^*} \right) = NW(p^*) \left( 1 - \frac{\|\pi^* - \bar{\pi}\|_\infty}{\pi_{min}^*} \right).$$

Rearranging the above conclusion and then applying the facts that $NW(p^*) \leq k/n$ (trivially) and $\pi_{min}^* \geq 1/n$ (Lemma 4.2), we get the desired additive bound:

$$NW(p^*) - NW(\bar{p}) \leq NW(p^*) \cdot \frac{\|\pi^* - \bar{\pi}\|_\infty}{\pi_{min}^*} \leq \frac{k}{n} \cdot \frac{\|\pi^* - \bar{\pi}\|_\infty}{1/n} \leq k \|\pi^* - \bar{\pi}\|_\infty \qquad\square$$

# D  Omitted Materials from Section 5

## D.1  Algorithm Descriptions

**Algorithms for calculating optimal panel distributions.**
In this paper, we calculate optimal panel distributions across instances with respect to Maximin, NW, and Leximin objectives. To do this, we build on publicly-available code [18], which implements the column generation techniques from [12].

**Rounding algorithms.**
At a high level, the task solved by the PIPAGE and BECK-FIALA rounding algorithms in Section 5

can be thought of as rounding an input panel distribution $p$ to some uniform lottery $\bar{p}$ by rounding the STANDARD LP described in Section 3. However, neither of these rounding methods are used to directly round $p$; rather, they are used to round a modified version $p'$, which transforms the task from rounding entries of $p$ to multiples of $1/m$ to the task of rounding entries of $p'$ to 0/1. The details of this transformation are described in the proof of Theorem 3.2 in Appendix B.

PIPAGE

We round $p'$ exactly according to the Pipage Rounding algorithm specified in Gandhi *et al* [16]. We note that their algorithm is specified for the task of rounding bipartite graphs; we apply their methods by formulating our rounding problem as a star graph, where each of the $|\mathcal{K}|$ vertices surrounding the central vertex corresponds to a feasible panel $P$. Each edge from the central vertex $i$ to a surrounding vertex $P$ has a weight (which will ultimately be rounded to 0/1) equal to $x_{i,P} = p'_P$, the probability of drawing panel $P$ from the modified version of the initial distribution $p'$. Gandhi *et al*'s degree preservation property guarantees the satisfaction of our adding up constraint $\|p'\| = \|\bar{p'}\|$.

BECK-FIALA

Our Beck-Fiala implementation is identical to the deterministic implementation specified in the proof of Lemma 9, Appendix B.4.1 of [13]. For details on the mapping of their setting to ours, see the proof of Theorem 3.2 in Appendix B.

**Integer Programs.**

IP-MAXIMIN

The below integer program computes a lottery $\bar{p} \in (\mathbb{Z}^+/m)^{|\mathcal{K}|}$, where the variables are $y$, the lower bound on any marginal probability; $\bar{p}$, the uniform lottery; and $\bar{\pi}$, the implied vector of marginals. The first constraint, along with the objective, result in the maximization of the minimum marginal. The second constraint imposes the relationship between the panel distribution $\bar{p}$ and the marginals $\bar{\pi}$. The third constraint imposes that the resulting panel distribution $x$ will be a uniform lottery. The fourth and fifth constraints impose that $\bar{p}$ is a valid distribution.

$$\text{Maximize } y$$
$$\text{s.t. } \bar{\pi}_i \geq y \qquad \forall i \in N$$
$$\sum_{\substack{P \in \mathcal{K}, \\ P \ni i}} \bar{p}_P = \bar{\pi}_i \qquad \forall i \in N$$
$$m\,\bar{p}_P \in \mathbb{Z}^+ \qquad \forall P \in \mathcal{K}$$
$$\sum_{P \in \mathcal{K}} \bar{p}_P = 1$$
$$\bar{p}_P \geq 0 \qquad \forall P \in \mathcal{K}$$

IP-NW

This integer program is essentially the same as IP-MAXIMIN, except that instead of maximizing the lower bound on the marginals, it maximizes the geometric mean of the marginals by equivalently maximizing the sum of their logarithms.

$$\text{Maximize } \sum_{i \in N} \log(\bar{\pi}_i)$$
$$\text{s.t. } \sum_{\substack{P \in \mathcal{K}, \\ P \ni i}} \bar{p}_P = \bar{\pi}_i \qquad \forall i \in N$$
$$m\,\bar{p}_P \in \mathbb{Z}^+ \qquad \forall P \in \mathcal{K}$$
$$\sum_{P \in \mathcal{K}} \bar{p}_P = 1$$
$$\bar{p}_P \geq 0 \qquad \forall P \in \mathcal{K}$$

IP-MARGINALS

This IP takes as input some panel distribution $p, \pi$ to be rounded, and minimizes the largest discrepancy of any resulting $\bar{\pi}_i$ from the corresponding $\pi_i$. Again, several of the constraints and variables

are common with IP-MAXIMIN.

$$\text{Minimize } z$$
$$\text{s.t. } |\pi_i - \bar{\pi}_i| \leq z \qquad \forall i \in N$$
$$\sum_{\substack{P \in \mathcal{K}, \\ P \ni i}} \bar{p}_P = \bar{\pi}_i \qquad \forall i \in N$$
$$m\,\bar{p}_P \in \mathbb{Z}^+ \qquad \forall P \in \mathcal{K}$$
$$\sum_{P \in \mathcal{K}} \bar{p}_P = 1$$
$$\bar{p}_P \geq 0 \qquad \forall P \in \mathcal{K}$$

### D.2  Implementation Notes and Algorithm Runtimes

Our experiments were implemented in Python and run on a 13-inch MacBook Air (2018) with a 1.6 GHz Intel Core i5 processor.

Runtimes of PIPAGE, BECK-FIALA, and IP-NW on rounding an unconstrained distribution are given in the table below. We optimized IP-NW with Gurobi using its built-in piecewise linear approximation of logarithms (given that IP-NW is nonlinear) with the parameter controlling the error in the piecewise approximation set to FuncPieceError=0.0001. This worked quite well in most instances, getting within $1/m$ of optimal fairness on 10 out of 11 instances.

IP-MAXIMIN and IP-MARGINALS were run in Gurobi and struggled to converge completely (even after many hours), but showed good performance after a short time. The results in the paper show their solutions after 30 minutes of run-time.

Table 2: Run-times for PIPAGE, BECK-FIALA, and IP-NW

| Instance | PIPAGE | BECK-FIALA | IP-NW |
|---|---|---|---|
| sf(a) | 1.5 | 1.6 | 17.1 |
| sf(b) | 1.3 | 1.3 | 27.8 |
| sf(c) | 1.0 | 1.1 | 33.1 |
| sf(d) | 2.1 | 2.3 | 40.6 |
| sf(e) | 17.0 | 28.3 | 7245* |
| cca | 4.4 | 6.4 | 7207* |
| hd | 1.5 | 1.7 | 120.1 |
| mass | 0.4 | 0.4 | 3.4 |
| nexus | 2.8 | 3.2 | 21.1 |
| obf | 2.3 | 2.4 | 22.3 |
| ndem | 2.2 | 2.6 | 34.8 |

* indicates capped at 7200s (2 hours). Time is measured in seconds. All times given (except those that timed out) represent the average over 3 runs.

### D.3  Analysis of Nash Welfare Fairness Preservation (Figure corresponding to Figure 2)

Here we give the corresponding analysis from Figure 2 for NW. We see, first that there is some algorithm in every instance that achieves within $0.1/m$ of $NW(p^*)$, where $p^*$ is the NW optimizing unconstrained distribution. This indicates that the cost of transparency to NW in practice is essentially 0. We note that in a few instances, IP-NW, which should theoretically dominate all other algorithms, is outperformed by either PIPAGE or BECK-FIALA. As we discuss in Appendix D.2, this is due to small errors in the integer optimization errors.

We find that our theoretical upper bounds on NW loss are less useful than those on the Maximin loss, because they are multiplied by an additional factor of $k$, while the value of the NW objective falls within a similar range to the Maximin objective. We note, however, that these bounds would be useful for larger $m$: currently, the maximum possible losses implied by the bounds fall between $191/m = 0.191$ and $5922/m = 5.922$. If we increased $m$ by a factor of 100 to $m = 100,000$ (this

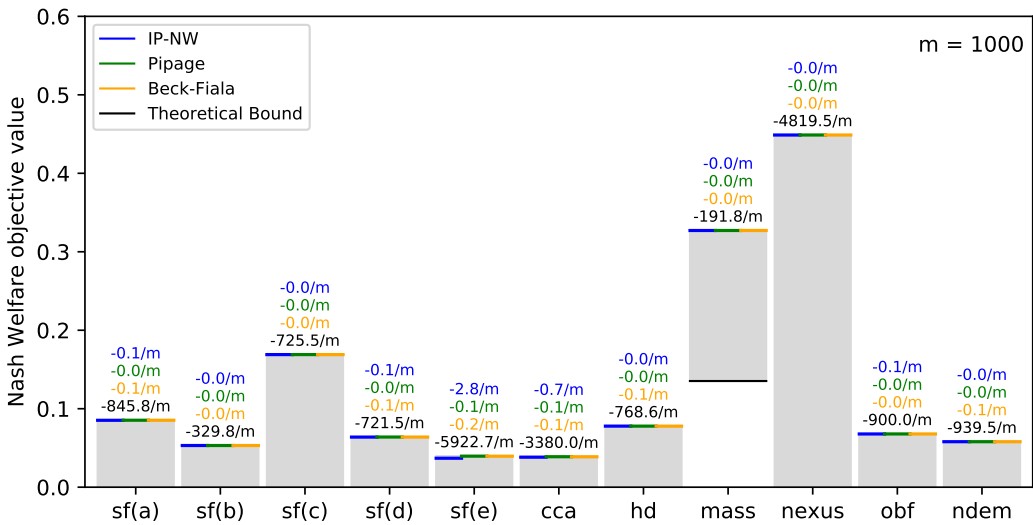

Figure 4: $m = 1000$. Shaded regions extend from $NW(p^*)$, the fairness of the optimal unconstrained distribution, down to the minimum fairness implied by the tightest theoretical upper bound in that instance (in all instances but "obf" Theorem 3.3 is tightest). Each algorithm or bound's loss relative to $NW(p^*)$ is written above in the corresponding color. We show a representative run of PIPAGE, a randomized algorithm.

would mean drawing 5 lottery balls instead of 3), then our bounds would be nearly tight to optimal in multiple instances (e.g., in "sf(a)", this would yield a loss of 0.008), and would be meaningful in all instances.

## D.4 Analysis of Leximin Preservation (Figures corresponding to Figure 3)

Here we give the corresponding analysis from Figure 3 for all other instances. In all instances, the conclusions we draw are essentially the same as those drawn from Figure 3: in all instances, all algorithms almost exactly preserve the Leximin-optimal marginals. Our theoretical bounds are meaningful, but we consistently outperform them in practice.

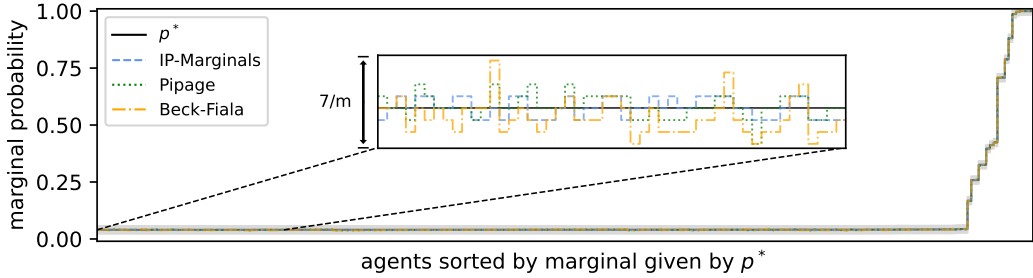

Figure 5: sf(b)

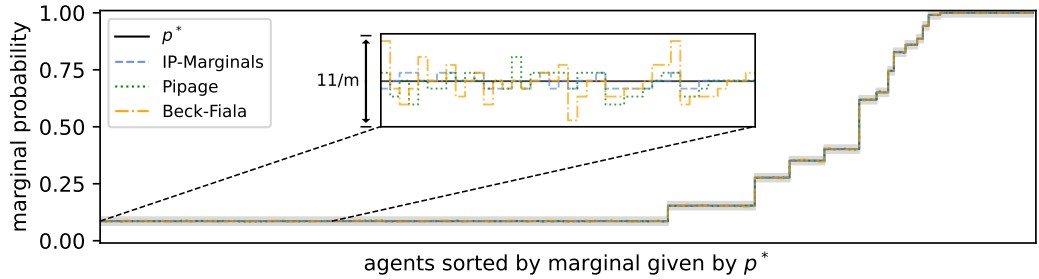

Figure 6: sf(c)

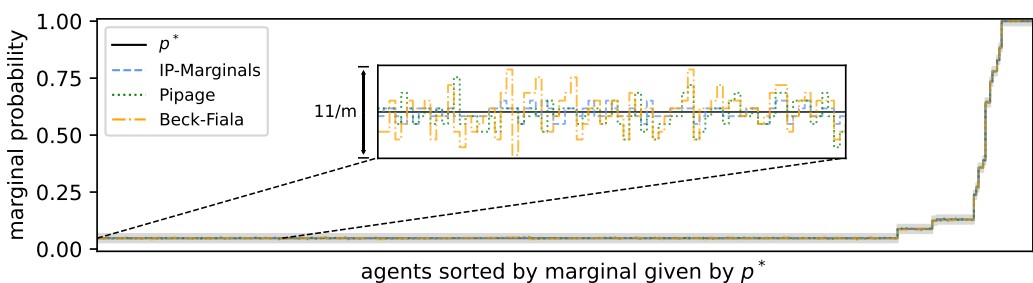

Figure 7: sf(d)

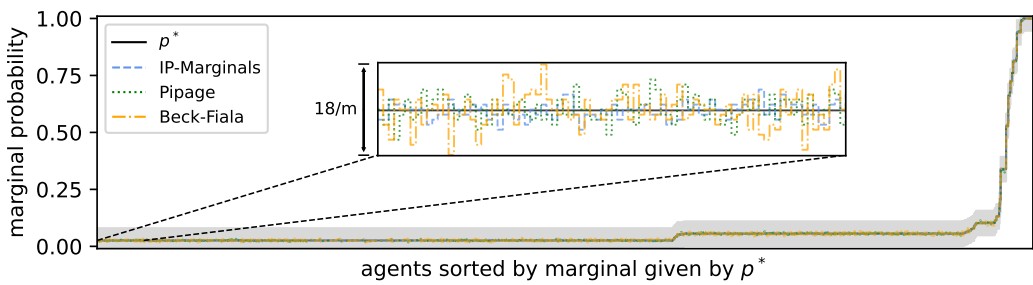

Figure 8: sf(e)

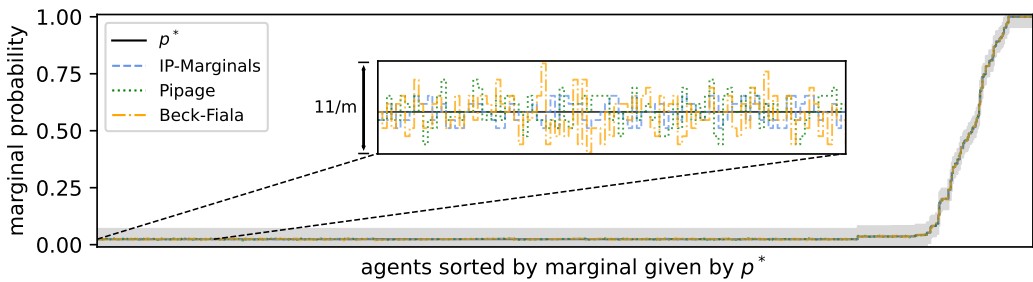

Figure 9: cca

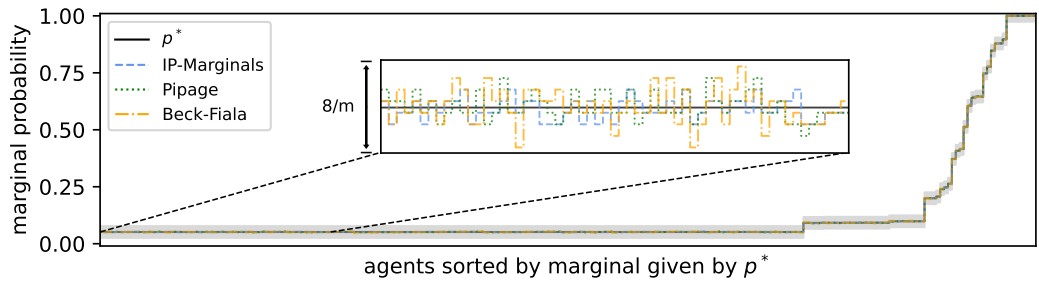

Figure 10: hd

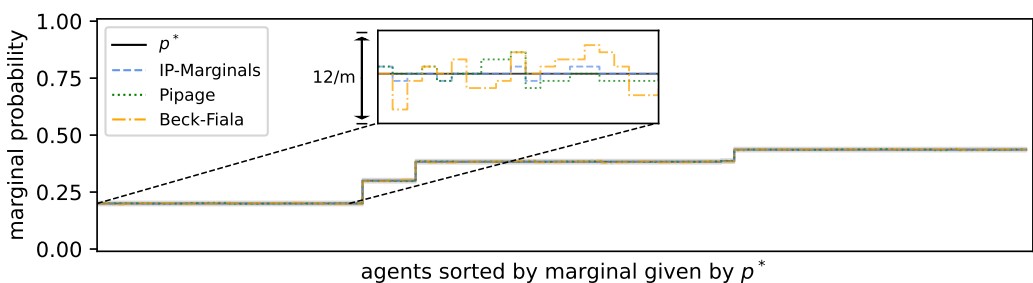

Figure 11: mass

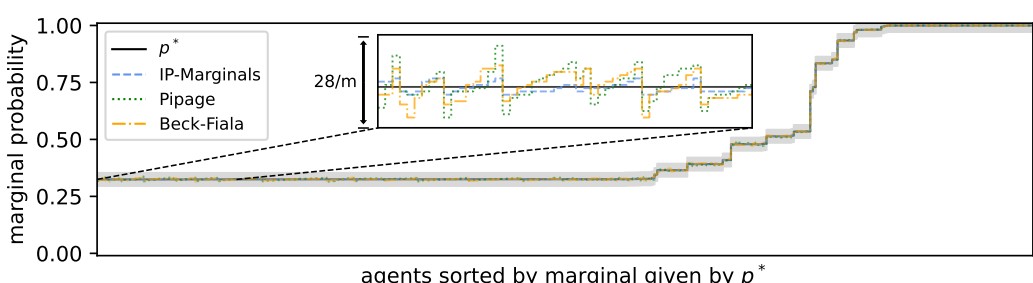

Figure 12: nexus

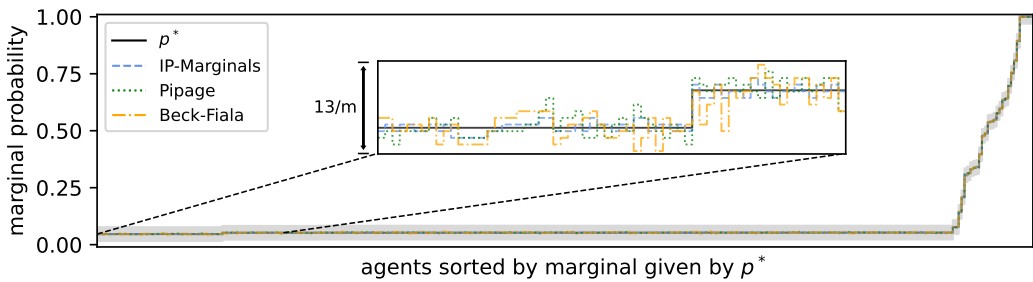

Figure 13: obf

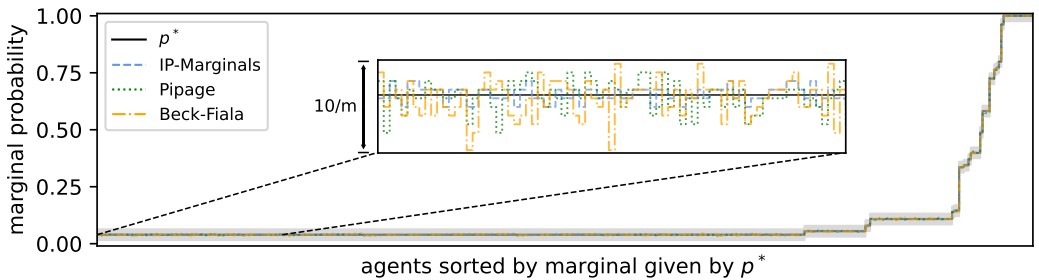

Figure 14: ndem