# OpenReview forum: "Fair Sortition Made Transparent"
_NeurIPS.cc/2021/Conference — NeurIPS 2021 Poster_

### Official Review · Reviewer_5XCt · 2021-07-04

**Rating:** 7
**Confidence:** 2

**Summary:**

This paper develops and characterizes methods for selecting a citizens' panel that is representative (satisfying upper and lower quotas on demographics' representation), fair (each eligible citizen has as equal a chance as possible of being selected), and transparent (where the final panel is chosen uniformly at random from a set of candidate panels in a public lottery).

**Limitations And Societal Impact:**

This paper is grounded in a read-world problem and is obviously concerned with the values of the people involved in and affected by sortition. The limitations of the proposed method (notably, that outside observers cannot distinguish between an individual's high selection probability being necessary for representation and being a way to rig the panel to include them) are clearly explained.

**Main Review:**

While I am not familiar with the sortition literature and did not check the mathematical details of this paper, I found it interesting and reasonably easy to follow. I'll let other reviewers to determine whether the problem studied (in particular, the transparency requirement) is important.

**Time Spent Reviewing:**

1

---

> ### Author Response · Authors · 2021-08-10
> **Response to Reviewer 4**
>
> We thank the reviewer for their comments.

---

### Official Review · Reviewer_jMTv · 2021-07-14

**Rating:** 7
**Confidence:** 3

**Summary:**

This paper gives a method for selecting panels via sortition that offers transparency that can be theoretically guaranteed and empirically audited. Their method is based on uniform lotteries and aims to achieve Maximin and Nash Welfare notions of fairness.

**Limitations And Societal Impact:**

The paper briefly acknowledges that the notion of transparency suggested in the paper looks at only the final stage of the panel selection process. I agree that thinking about transparency throughout the selection process is important from recruitment to final selection.

Another major consideration is that quotas may not be legally compatible in certain domains. Transparency may lead to legal challenges that violate equal protection laws (particularly in the United States). What does this mean for comparing against quota systems of sortition-based selection or stratified sampling?


**Main Review:**

While previous works present the merits of sortition algorithms and give methods to better sample for sortition based committee selection, this work furthers these goals by analyzing uniform lottery with fairness constraints.

The worst-case marginal discrepancy bounds seem reasonable and it is great to see a tighter upper bound for n >> k^2 case since that is likely applicable in most applications.

I am not sure I fully understand the argument of why a worst-case Maximin guarantee is a Leximin guarantee. It seems like uniform lottery does not necessarily preclude Leximin solutions.

It is nice that the authors consider both Maximin and Nash Welfare when bounding the fairness difference between the p* and \bar{p}. What explains the extra factor of k between the Maximin and Nash upper bounds?

Considering that sortition is increasingly used in the real world, I believe this paper makes an important contribution in analyzing fairness costs of a transparent sortition process: uniform lottery.


**Time Spent Reviewing:**

3

---

> ### Author Response · Authors · 2021-08-10
> **Response to Reviewer 3**
>
> *1. I am not sure I fully understand the argument of why a worst-case Maximin guarantee is a Leximin guarantee. It seems like uniform lottery does not necessarily preclude Leximin solutions.*
>
> R3 is correct that in general, a Maximin guarantee is not equivalent to a Leximin guarantee (in fact, as we discuss in the paper, there is not an obvious way to write down a "Leximin guarantee" as a single mathematical expression). However, a bound on Maximin loss is informative about the *highest-priority* type of loss according to Leximin: recall that Leximin and Maximin both care most about maximizing the minimum marginal: it is Leximin's *first* priority, while it is Maximin's *only* priority. Thus, in the multitude of instances where discretizing a distribution to multiples of $1/m$ requires decreasing minimum marginal probability, what we care about most -- regardless of whether the objective we aim for is Leximin or Maximin -- is upper-bounding the required loss in the *minimum probability*, which is exactly the bound given on the Maximin loss. We will make this phrasing clearer in the text.
>
> *2. It is nice that the authors consider both Maximin and Nash Welfare when bounding the fairness difference between the $p^\star$ and $\bar{p}$. What explains the extra factor of $k$ between the Maximin and Nash upper bounds?*
>
>
> First, Figure 1 and Figure 3 suggest that in practice, Nash Welfare (NW) actually decreases less than Maximin after rounding, so it is unclear whether the additional factor of $k$ between Corollary 4.1 and Theorem 4.2 is inherent or simply an artifact of the proof. In particular, it seems unlikely that Theorem 4.2 is tight.
>
> In the analysis in Theorem 4.2, this extra factor of $k$ arises because Maximin depends additively on the change in the smallest marginal, while NW depends *multiplicatively* on this change (put another way, small additive losses in individuals' marginal can have outsize impacts on Nash Welfare if they are applied to already-small marginals, as their multiplicative effect on small marginals will be large). Since we show that the minimum marginal in the unrounded NW-optimal distribution is at least $1/n$, while the average marginal is $k/n$, the NW is at most a factor of $k$ times "more sensitive" to changes in the minimum marginal at optimality than it would be to a change in the average marginal. We then use this as an upper bound on its sensitivity to any marginal, and so our final guarantee inherits this same factor of $k$.
>
> *3. (Limitations and Societal Impact) Another major consideration is that quotas may not be legally compatible in certain domains. Transparency may lead to legal challenges that violate equal protection laws (particularly in the United States). What does this mean for comparing against quota systems of sortition-based selection or stratified sampling?*
>
> While there are many settings in society in which explicit demographic quotas are not permitted, citizens' assemblies are not currently one of them; the use of quotas for selecting assemblies is a widely accepted practice across many countries, including the US [FGG+21].
>
> We've also spoken with several of the practitioner organizations (including multiple that have run panels in the US), and none have expressed concerns about potential legal issues over the mathematical impossibility of making selection probabilities exactly equal. In fact, the lottery method we propose in this paper was used to select a panel in Michigan in 2020, and there were no legal issues raised. In the future, if this becomes a concern, there are ways to address the lack of equality of probabilities: one could decrease the skew of the pool toward certain demographics (possibly by offering stronger incentives for participation), and/or loosen the quotas.
>
> We note that stratified sampling requires the use of quotas, so if quotas are not legally viable, then stratified sampling is also likely not legally viable. For more information about how the orthogonal quotas we impose with our methods are strictly more general than the quotas that can be imposed by stratified sampling, see Appendix 3 of Flanigan *et al.* [FGG+21].
>
> Citations: [FGG+21] Flanigan, Bailey, et al. "Fair algorithms for selecting citizens’ assemblies." Nature (2021): 1-5.

---

### Official Review · Reviewer_tP62 · 2021-07-16

**Rating:** 7
**Confidence:** 2

**Summary:**

Additional to the maximally-fair consideration, this paper concerns the transparency in assembly selection problem. Specifically, the paper introduces the notion of transparency in panel selection as people can easy understand the probabilities with which each individual will be chosen for the panel and verify that individuals are actually selected with these probabilities. To achieve this, the paper studies the $m$-uniform lottery where the selection probability of each feasible panel must be multiples of $1/m$.
The paper shows that there exists a uniform lottery over $m$ panels that can nearly preserve the fairness of the maximally-fair unconstrained distribution over panels.

Furthermore, the paper uses the fairness loss and marginal discrepancy to quantify the closeness of such lottery and the lottery obtained in an unconstrained (no transparency requirement) setting. The paper characterizes several upper bounds for these measures and further strengthen these bounds in more structured settings.

Lastly, the paper conducts experiments on real-world panel selection instances to demonstrate the viability of the uniform lottery approach as a method of selecting assemblies both fairly and transparently.



**Limitations And Societal Impact:**

Yes

**Main Review:**

Originality, quality, clarity, and significance

I should first mention that I feel I'm not a suitable reviewer for this paper, as the main topics are outside my areas of expertise – I have indicated this in my low confidence for my score. Although I cannot judge its novelty adequately, the problem studied in this paper is well motivated and exhibits solid applications in real-world. The theoretically results around the problem setting appear correct and the experiments seem to be comprehensive.
In terms of the presentation, the paper is overall well written. I would like to appreciate author's efforts in introduction section to make the motivation of the problem well defined.

Questions and comments:

1. are there any connections between $|\mathcal{K}|$ and $m$? I'm a bit confusing here, as in Line 90, you mentioned "uniform lottery over $m$ panels", and $\mathcal{K}$ is the set of all feasible panels by definition in Line 129.

2. I found it a bit difficult to parse the definition of Maximin, it might simply because that I'm not in this area. Seems that Maximin can not bound the difference of marginal of individuals, which is usually the focus of individual fairness. Could author explain this a little bit?

3. The fairness objectives considered in the paper are more from the point of view of individuals. I'm wondering whether the current results still hold if you replace the objectives to a group fairness measure?

4. Does $k\le m$ hold for sure? If $k>m$, Theorem 3.2 seems to be meaningless?


Others:

Line 248: "be most natural start with" -> starting

Line 281, you probably want to unify Nash-Welfare optimal and NW-optimal.

Line 291, it should be "Claim B.6"?

==== Post Rebuttal =====

Thank the authors for the clear and detailed feedback. Taking everything into consideration, I'd like to keep my rating as-is.

**Time Spent Reviewing:**

20

---

> ### Author Response · Authors · 2021-08-10
> **Response to Reviewer 2**
>
> We thank the reviewer for their comments, and we will make the appropriate line edits and clarifications.
>
> *1. Are there any connections between $|{\cal K}|$ and $m$? I'm a bit confused here, as in Line 90, you mentioned "uniform lottery over panels" and $|{\cal K}|$ is the set of all feasible panels by definition in Line 129.*
>
> ${\cal K}$ is the collection of all unique panels that satisfy the quotas in a given instance, and thus is determined by the pool, the quotas, and the panel size; $m$ is an unrelated parameter chosen by the practitioners running the panel, representing the number of panels, possibly containing duplicates, over which they ultimately want to randomize using a physical lottery. $m$ would be chosen by practitioners not dependent on the quotas or on ${\cal K}$, but rather according to the practitioner's preferences about the size of their lottery, subject to the (in practice, very weak) requirement that $m$ is large enough to avoid meaningful losses in fairness. Although these terms are theoretically not connected, in practice we expect that $|{\cal K}|\gg m$, which is useful to ensure that a lottery of size $m$ can achieve fairness to individuals.
>
> The "uniform lottery over panels" we refer to is over the chosen number of (not necessarily unique) panels $m$ where all panels in this uniform lottery are in the set ${\cal K}$. *Uniform* here refers to the fact that all $m$ panels in this multi-subset are chosen with equal probability $1/m$. This uniform lottery is equivalently represented as a probability distribution over $\cal K$ for which every probability is a multiple of $1/m$: For example, if a panel received probability $7/m$ in the discretized distribution, there would be 7 copies of that panel among the $m$ panels from which we uniformly draw. We will try to better clarify the meaning of  "uniform lottery over panels" in the paper.
>
>
> *2. I found it a bit difficult to parse the definition of Maximin, it might simply because I'm not in this area. Seems that Maximin can not bound the difference of marginal of individuals, which is usually the focus of individual fairness. Could author explain this a little bit?*
>
> (By "difference of marginal of individuals," we take R2 to be referring to $\max_i \pi_i - \min_i \pi_i$, the difference between the smallest and largest marginals given to individuals induced by a panel distribution $p$.)
>
> R2 is correct that Maximin is distinct from their proposed fairness objective: Maximin is expressed as ${\max_{p} (\min_i \pi_i)}$ (maximizing the minimum marginal), whereas this proposed objective is ${\min_{p} (\max_i \pi_i - \min_i \pi_i)}$ (minimizing the maximum distance between any two marginals).
>
> We focus on Maximin because it is a simpler version of the *Leximin* objective, which is the well-established fairness objective that is already being maximized by existing panel selection algorithms used in practice [FGG+21]. In particular, the two objectives are related as follows: Leximin first runs Maximin (maximizes the minimum marginal), then subject to no probability being lower than that, maximizes the next smallest marginal, and so on. Therefore, a bound on the loss in Maximin fairness provides a bound on the most important loss in Leximin (whose loss cannot be expressed as a neat expression).
>
> That said, the loss in the fairness objective proposed by R2 due to rounding can also be upper-bounded as twice the magnitude of the loss in the Maximin objective with a simple extension of our bounds on the $\ell_\infty$ change in marginals due to rounding.
>
>
> *3. The fairness objectives considered in the paper are more from the point of view of individuals. I'm wondering whether the current results still hold if you replace the objectives to a group fairness measure?*
>
> Our guarantees for individuals directly imply guarantees for groups. In particular, we upper-bound the change in marginals $|\pi_i - \bar{\pi}_i|$ for all agents $i$ simultaneously in Theorems 3.1 and 3.2, and these bounds imply an upper bound on the marginal change for an arbitrary group which grows linearly in the group's size. By making $m$ large enough (and thus making the upper-bound on $|\pi_i - \bar{\pi}_i|$ small enough) relative to the size of any group, one can get bounds on losses in group fairness that improve in $m$.
>
> Of course, one might want stronger guarantees, and for getting such guarantees, we believe that our general strategy would remain a sound approach: to first (1) upper-bound the sensitivity of a fairness objective to changes in *group* marginals, i.e.,  the expected numbers of individuals selected from each given group (rather than individual marginals, as we do in this paper); and then (2) derive a rounded distribution over panels which approximately preserves all group marginals.
>
> When considering how our results extend to stronger bounds on group fairness, it is worth distinguishing between collections of groups which are defined in terms of the feature values defining the quotas, and collections of groups which are not.
>
> First considering groups given by feature values used to define the quotas: the group marginals for groups defined by a *single* feature value (e.g., women) are guaranteed deterministically by the quotas, which enforce that their group must receive some number of seats (possibly within a pre-defined range). For groups defined by more than one feature value simultaneously (e.g., young women) the approach which we present in Section 3.2, where we project the distribution into panel type space and then round it, uses discrepancy in a way that naturally approximately preserves group marginals for the groups defined by each feature vector present in the pool (and thus defined by any subset of feature values). Thus, our approach already gives some upper-bound on the change in group marginals, which could translate to upper-bounds on fairness loss, depending on the choice of fairness objective.
>
> It is less clear how to derive tight upper bounds on group marginals for other collections of groups. When the number of arbitrary groups is small, randomized rounding may give the best simultaneous guarantee on all group marginals.
>
>
> *4. Does $k \leq m$ hold for sure? If $k > m$, Theorem 3.2 seems to be meaningless?*
>
>
> Indeed Theorem 3.2, along with many of our other results, is not meaningful for $k > m$. For sortition in practice it is always the case that $k \leq m$, for the reasons described below.
>
> Practitioners get to choose $m$, and it is relatively easy and desirable to choose $m$ to be much larger than $k$. This is first because $k$, the size of the panel is generally quite small (typically between 30 - 100), and its size is limited logistically by the need for the panel to deliberate. In contrast to $k$, $m$ can be arbitrarily large, and each additional factor of 10 in the size of $m$ requires drawing only one more lottery ball (and the larger $m$ is, the stronger our fairness guarantees are). To have an interesting lottery, one probably needs to draw at least 3 balls -- i.e., $m = 1000$ panels -- already well exceeding practical values of $k$.
>
> Citations: [FGG+21] Flanigan, Bailey, et al. "Fair algorithms for selecting citizens’ assemblies." Nature (2021): 1-5.

---

### Official Review · Reviewer_oxDH · 2021-07-17

**Rating:** 6
**Confidence:** 4

**Summary:**

The paper discusses transparency in the context of sortition - selecting citizens to participate in a “Citizens’ Assembly” for the purpose of making policy recommendations. They seek to design algorithms which are approximately fair (with respect to three metrics - Maximin, Leximin, and NW) while being more interpretable/verifiable than disclosing selection probabilities. Their proposed solution is to design a uniform lottery, which selects a panel uniformly at random from a precomputed set of panels. They bound the fairness loss and deviation in marginals due to restricting to uniform lotteries, discuss LP rounding algorithms to find such lotteries, and run experiments on 11 real-world instances comparing the algorithms.

**Ethical Concerns:**

Privacy is a potential concern, as was alluded to briefly in line 68. Anonymization of pool members is mentioned (presumably for the sake of privacy), but there is no discussion of the extent of recoverability of personal information.

**Limitations And Societal Impact:**

Societal impact and limitations are adequately addressed.

**Main Review:**

The paper is well-written, clearly explaining the problem and techniques used. The theoretical results are basically applications of known randomized rounding algorithms over a natural linear program for this problem. The empirical results seem to indicate that imposing their definition of transparency is not harmful in terms of performance, and their methods seem implementable in practice. I liked the premise of the paper to make randomization more "transparent" by preselecting a set of $m$ panels, each of which could be a feasible panel, and then providing a uniform lottery on these to make the process look transparent. The main drawback to the paper is that the theoretical contribution is small—the LP rounding techniques used are not quite original.

- I wonder though, if the set of m panels that are computed could be modified in a way to actually bias any uniformly selected panel. What's the guarantee that the $m$ panels selected are all possible panels? In fact, moving the upper and lower bounds slightly could have a significant impact on the kind of panels obtained by uniformly sampling the feasible set.

**Time Spent Reviewing:**

1 (by me), 3 (by my student)

---

> ### Author Response · Authors · 2021-08-10
> **Response to Reviewer 1**
>
> *1. I wonder though, if the set of m panels that are computed could be modified in a way to actually bias any uniformly selected panel. What's the guarantee that the  panels selected are all possible panels? In fact, moving the upper and lower bounds slightly could have a significant impact on the kind of panels obtained by uniformly sampling the feasible set.*
>
> We start by clarifying a few things about our approach to constructing a uniform lottery over $m$ panels.
>
> First, our approach to building a uniform lottery *does not* consist of adjusting the quotas until exactly $m$ quota-satisfying panels exist, and then uniformly sampling these panels. In fact, the rounding procedures we propose do not adjust the quotas at all (leaving no opportunity for them to introduce bias via quota adjustments); rather, our methods take the quotas as a fixed feature of the instance we are given. In practice, the quotas are fixed by practitioners long before our proposed rounding procedure would be used.
>
> To build a nearly maximally fair uniform lottery over $m$ panels, we take as input a maximally fair distribution over ${\cal K}$, the set of all quota-satisfying panels. This maximally fair distribution (optimized according to Maximin, Nash, or one of several other fairness objectives) can be computed by the algorithms of Flanigan *et al* [FGG+21]. Our methods in this paper show how to then construct a *discretized* version of this input distribution, where a discretized distribution is one whose probabilities are multiples of $1/m$. (Since $m\ll |\cal K|$, most feasible panels would have probability 0.) Ideally, this discretized distribution closely preserves the fairness achieved by the original unrounded distribution (whether or not such a rounding always exists is one of the main theoretical questions answered in this paper). Note that the discretized distributions computed via our various rounding methods are guaranteed to be over exclusively *quota-satisfying* panels, as we show in the descriptions and proofs of our rounding processes. Sampling a final panel from such a discretized distribution can then be manifested as the *uniform* sampling a set of $m$ panels, possibly containing duplicates: for example, if a panel received probability $7/m$ in the discretized distribution, there would be 7 copies of that panel among the $m$ panels from which we uniformly draw. We will go through the paper and try to make these points clearer.
>
> *2. (Ethical Concerns). Anonymization of pool members is mentioned (presumably for the sake of privacy), but there is no discussion of the extent of recoverability of personal information.*
>
> In practice, there is little to no risk of individuals' information being recoverable when selecting panels via a uniform lottery, even after the selected panel is revealed. During selection by lottery, the only outward-facing information is the panel compositions, in which pool members are represented as numbers; viewers know the selection probabilities associated with these numerical identifiers, but know nothing about the demographic composition of the pool, and often don't even know the precise quotas. The one situation in which a small amount of individuals' information could be discerned by an observer would be if there were pool members who had to be chosen deterministically to satisfy a quota. Then, if one knew the quotas---and the quotas were sufficiently disparate in magnitude (which they often aren't)---someone might be able to guess *one* of these anonymized individuals' features.
>
> Citations:
> [FGG+21] Flanigan, Bailey, et al. "Fair algorithms for selecting citizens’ assemblies." Nature (2021): 1-5.

---

### Decision · Program_Chairs · 2021-09-27

**Decision:**

Accept (Poster)

**Comment:**

Thanks for the strong submission; the reviewers unanimously enjoyed it.